



# Coupled modelling of hydrological processes and grassland production in two contrasting climates

[1*]Nicholas Jarvis, [2,3]Jannis Groh, [1]Elisabet Lewan, [1]Katharina H. E. Meurer, [4]Walter Durka, [4]Cornelia Baessler[4], [2]Thomas Pütz, [1]Elvin Rufullayev, [2]Harry Vereecken

[1]*Soil and Environment, Swedish University of Agricultural Sciences, Uppsala, Sweden*
[2]*Agrosphere (IBG-3), Institute of Bio- and Geoscience, Forschungszentrum Jülich GmbH, Jülich, Germany*
[3]*Research Area 1 "Landscape Functioning", Working Group "Hydropedology", Leibniz Centre for Agricultural Landscape Research (ZALF), Müncheberg, Germany*
[4]*Department of Community Ecology (BZF), Helmholtz Centre for Environmental Research (UFZ), Halle, Germany*

*corresponding author (nicholas.jarvis@slu.se)





**Abstract**
Projections of global climate models suggest that ongoing human-induced climate change will
lead to an increase in the frequency of severe droughts in many important agricultural regions
of the world. Eco-hydrological models that integrate current understanding of the interacting
processes governing soil water balance and plant growth may be useful tools to predict the
impacts of climate change on crop production. However, the validation status of these models
for making predictions under climate change is still unclear, since few suitable datasets are
available for model testing. One promising approach is to test models using data obtained in
"space-for-time" substitution experiments, in which samples are transferred among locations
with contrasting current climates in order to mimic future climatic conditions. An important
advantage of this approach is that the soil type is the same, so that differences in soil
properties are not confounded with the influence of climate on water balance and crop
growth. In this study, we evaluate the capability of a relatively simple eco-hydrological model
to reproduce 6 years (2013-2018) of measurements of soil water contents, water balance
components and grass production made in weighing lysimeters located at two sites within the
TERENO-SoilCan network in Germany. Three lysimeters are located at an upland site at
Rollesbroich with a cool, wet climate, while three others had been moved from Rollesbroich
to a warmer and drier climate on the lower Rhine valley floodplain at Selhausen. Four of the
most sensitive parameters in the model were treated as uncertain within the framework of
the GLUE (Generalized Likelihood Uncertainty Estimation) methodology, while the remaining
parameters in the model were set according to site measurements or data in the literature.
The model accurately reproduced the measurements at both sites, and some significant
differences in the posterior ranges of the four uncertain parameters were found. In particular,
the results indicated greater stomatal conductance as well an increase in dry matter allocation
below-ground and a significantly larger maximum root depth for the three lysimeters that had
been moved to Selhausen. As a consequence, the apparent water use efficiency (above-
ground harvest divided by evapotranspiration) was significantly smaller at Selhausen than
Rollesbroich. Data on species abundance on the lysimeters provide one possible explanation
for the differences in the plant traits at the two sites derived from model calibration. These
observations showed that the plant community at Selhausen had changed significantly in
response to the drier climate, with a significant decrease in the abundance of herbs and an
increase in the proportion of grass species. The differences in root depth and leaf conductance
may also be a consequence of plasticity or acclimation at the species level. Regardless of the
reason, we may conclude that such adaptations introduce significant additional uncertainties
into model predictions of water balance and plant growth in response to climate change.



## 1. Introduction

Projections of global climate models suggest that ongoing human-induced climate change will lead to an increase in the frequency of severe droughts (Ruane et al., 2018). This may seriously impact production in many important agricultural regions of the world (Tubiello et al., 2007), including managed grasslands (e.g. Kipling et al., 2016; Stanimirova et al., 2019), since key forage species are known to be sensitive to drought (Norris, 1982; Coleman et al., 1989; Silvertown et al., 1994; Jenkinson et al., 1994; Volaire et al., 1998; Meurer et al., 2019). Grasslands are also of major importance in the context of climate change mitigation, since they cover ca. 70% of the global agricultural land area (Foley et al., 2011) and represent a large store of soil organic carbon (SOC) (Li et al., 2018; Bossio et al., 2020). Soil water status affects plant growth through a complex web of direct and indirect mechanisms (Körner, 2015; White et al., 2016; Tardieu et al., 2018; Loka et al., 2019; Gupta et al., 2020). In turn, plant growth, both above- and below-ground, influences the soil water balance through important feedback mechanisms, particularly the regulation of transpiration by leaf area as well as the control of water supply from the soil by root length density and its distribution with depth (Monteith, 1986, 1988; Tardieu et al., 2017). Thus, realistic models of the coupled processes of root water uptake, transpiration and plant growth are required to predict reliably the impacts of climate change on the future productive potential of grassland. Eco-hydrological models that attempt to capture these interactions in the soil-plant system are widely used in climate change studies that focus on the prediction of latent and sensible heat fluxes and $CO_2$ exchange between the land surface and the atmosphere (e.g. Fatichi et al., 2016; Klein et al., 2017; Kellner et al., 2017). Similarly, soil-crop models that integrate current understanding of the interacting processes governing water balance, SOC and nutrient cycling and crop growth (e.g. Robertson et al., 2015; Wu et al., 2016; Stöckle and Kemanian, 2020) are often used as tools to predict the impacts of land use or climate change on crop production and the environment (e.g. Eckersten et al., 2012). These two types of simulation model share many similarities. In the following, we refer to them collectively as SVAT (soil-vegetation-atmosphere) models.

SVAT models employ empirical (or phenomenological) approaches to describe many of the key processes in the soil-plant system. This is especially the case for the processes governing plant growth because the underlying mechanisms are extremely complex and not easily amenable to fundamental descriptions (Boote et al., 2013; Wu et al., 2016). This means that great care is needed in model calibration exercises, given the usual paucity of experimental data in relation to the number of model parameters. In such cases, parameter errors may often compensate for model deficiencies leading to non-unique solutions or 'equifinality' (Beven and Binley, 1992; Beven, 2006). Parameter uncertainty has not always been considered in SVAT model applications (Seidel et al., 2018). Thus, even though a model performs satisfactorily, it may be doing so for the wrong reasons (Kirchner, 2006). As a consequence, model predictions, for example for a future climate, can be seriously in error (Kersebaum et al., 2007, 2015; Bellocchi et al., 2010; He et al., 2017). In this respect, despite their great potential, it is not yet clear how accurately SVAT models can predict the soil water balance and production potential of grasslands in a changing climate because few suitably comprehensive data sets have been available to unequivocally constrain them in model calibration exercises. Several SVAT models specifically designed for grassland agro-ecosystems





have been developed (e.g. Jouven et al., 2006a,b; Jing et al., 2012; Persson et al., 2014).
However, with only a few exceptions, previous studies have focused on calibrating these
models against data on above-ground biomass production at single sites, with scant focus on
hydrological processes and below-ground biomass, and with little attention paid to parameter
uncertainty. In a test of the *PaSim* grassland model at the regional scale, Ma et al. (2015) found
that although $CO_2$ and water fluxes between the land surface and atmosphere were
reasonably well matched, soil water contents were not accurately simulated during dry
periods. Similarly, in a multi-model and multi-site validation exercise, Sándor et al. (2017)
noted a variable model performance at sites with contrasting climates. In particular, they
demonstrated a failure of the models to simulate correctly root water uptake patterns and
biomass production in dry summers and at dry sites. Even though most grassland species are
generally comparatively shallow-rooted (Jackson et al., 1996), several previous studies have
highlighted the role of sparsely distributed deeper roots in maintaining water uptake,
transpiration and growth during droughts (e.g. Kemp and Culvenor, 1994; Volaire et al 1998;
Bonos and Murphy, 1999; Zwicke et al., 2015). This suggests that models of root water uptake
for grass must account for compensatory mechanisms, whereby water uptake increases from
sparsely rooted wetter soil layers to compensate for reductions in water uptake in densely
rooted, but dry soil (Jarvis, 2011; Cai et al., 2017).
Manipulation experiments have been carried out to simulate the effects of climate change on
grasslands in which plant growth has been monitored following controlled alterations in the
precipitation regime (e.g. reduced rainfall amount or frequency). However, nearly all of these
experiments are of a short-term nature and the treatments imposed have often been extreme
and thus not well adapted to climate model projections (e.g. Beier et al., 2012; Hoover et al.,
2018). Furthermore, with only a few exceptions (e.g. Bollig and Feller, 2014), drought
manipulation experiments have not focused much on the complex interactions between soil
hydrological processes, water stress and plant growth, despite their importance. Thus, in most
cases, the mechanisms controlling the observed growth responses have not been elucidated
in detail, while little data is available from these experiments that could support and test
model predictions (Beier et al., 2012; Hoover et al., 2018). An alternative approach is to test
model performance against data obtained in "space-for-time" substitution experiments, in
which samples are transferred among sites with contrasting current climates in order to
approximately mimic likely future climate conditions (Ineson et al., 1998; Pütz et al., 2016).
One important advantage of this approach is that the soil type is the same, so that differences
in soil properties are not confounded with the influence of climate on soil hydrology and crop
growth. Weighing lysimeters are highly suitable study objects in this context, since they enable
the measurement of a complete (closed) water balance (Wegehenkel et al., 2008; Heinlein et
al., 2017; Groh et al., 2020a). Providing they are sufficiently large in terms of both depth and
diameter, weighing lysimeters also represent a relatively natural environment for plant
growth as well as allowing the installation of instrumentation to measure soil water status.
In this study, we make use of data from the TERENO-SoilCan network, in which large weighing
lysimeters containing undisturbed soil monoliths have been transferred among several
locations in Germany to emulate expected changes in climate (Zacharias et al., 2011; Pütz et
al., 2016; Groh et al., 2020b). In this study, we evaluate the capability of a relatively simple





eco-hydrological model to reproduce six years of measurements of the soil water balance and
grassland production in lysimeters at two sites in the Eifel/Lower Rhine Valley observatory
(Zacharias et al., 2011; Pütz et al., 2016; Bogena et al., 2018). Three of these lysimeters are
located at an upland site at Rollesbroich with a cool, wet climate, while three others were
moved from Rollesbroich to a warmer, drier climate in the Rhine valley at Selhausen.

## 2. Materials and methods

### 2.1 Site descriptions, vegetation, soil properties and lysimeter data

The station at Rollesbroich (50º 37' N, 6º 18' E) is located on a hilltop site at an elevation of
511 m, while Selhausen (50º 52' N, 6º 27' E) is located on a relatively flat alluvial flood plain in
the lower Rhine valley at an altitude of 104 m. The mean annual air temperature at
Rollesbroich is 8ºC and the mean annual precipitation is 1150 mm. At Selhausen, the mean
annual air temperature is 10ºC and the mean annual precipitation is 720 mm. A weather
station at each site records precipitation, solar radiation, air temperature, air humidity and
wind speed at a height of 2 m at a ten-minute time resolution (Pütz et al., 2016), which we
aggregated to a daily time step. From these meteorological variables, we calculated daily
reference (potential) evapotranspiration for grass with the FAO Penman-Monteith equation
(Allen et al., 1998) as a simple comparative measure of the atmospheric demand for water in
the two climates. The meteorological data and calculated reference evapotranspiration at the
two sites for the period 2013-2018 are shown in the supplementary information (figure S1).
The soil at Rollesbroich is a Stagnic Cambisol, with the basic properties shown in Table 1. The
soil is a sandy loam in the topsoil, changing abruptly to a clay loam at 24 cm depth. The texture
again becomes coarser (sandy loam/loam) in the deep subsoil below 93 cm (Table 1). The
original grassland community on the lysimeters extracted at Rollesbroich is classified as a
mesic grassland of the Arrhenatheretalia alliance without any clear affiliation to classical plant
associations. The community is dominated by *Lolium perenne* L., *Ranunculus repens* L., *Rumex*
*acetosa* L., *Taraxacum officinale* L., *Dactylis glomerata* L. and *Trifolium repens* L. During the
extraction of the lysimeters at Rollesbroich, grassland roots were observed to extend to ca.
40-50 cm depth (J. Groh, T. Pütz, pers. comm.). This is supported by SOC contents measured
in the soil profile, which decline abruptly below 50 cm depth (Table 1). The lysimeters are
supplied with fertilizer as liquid manure and the vegetation is cut 3 to 4 times per growing
season to characterize above-ground biomass production, following the local management
practice. During the first four years (2013-2016) of the experimental period, leaf area index
was measured on multiple occasions with an LAI-2200C Plant Canopy Analyzer from Licor.
Plant height was also measured using a conventional ruler. Plant communities present in the
lysimeters were assessed annually during the period 2011 to 2016. Plant species abundance
was estimated as the number of grid cells occupied of 64 rectangular cells (10 × 10cm). Based
on this data, the relative abundances of three plant functional types (i.e. grasses, legumes and
non-legume herbs) were quantified. These observations showed that plant communities
changed significantly at both sites, with a general decrease in the abundance of herbs and an
increase in the proportion of grass species (figure S2). This change was much less pronounced
at Rollesbroich than in the lysimeters transferred to Selhausen, where the plant community



composition diverged continuously from the original resident community composition,
presumably in response to the move to the warmer and drier climate. The small changes in
community composition found at Rollesbroich may be a consequence of the experimental set-
up. For example, the lysimeters do not allow for root-ingrowth of rhizomatous herb species.
The lysimeters have a surface area of 1 m$^2$ and are 1.5 m deep. Weighing devices (load cells)
measure weight changes equivalent to a water depth of 0.01 mm. Application of a filter
routine to separate signal from noise enables accurate estimations of both precipitation and
evapotranspiration from each lysimeter (Peters et al., 2017). Missing precipitation data were
filled in a first step using the mean value calculated for all available lysimeters. In a second
step, any remaining gaps were then filled using the precipitation measured by the reference
precipitation gauge. Water fluxes into and out of the lysimeters at the base are controlled by
measurements of soil water pressure heads made in the surrounding soil at 1.4 m depth. Soil
water contents and pressure heads are measured at a ten-minute time resolution at three
depths (10, 30 and 50 cm depth) in the lysimeters using TDR probes and tensiometers (30 and
50 cm depth) or matric potential sensors (10 cm depth). A detailed description of the design,
construction and extraction of the lysimeters and their installation in the lysimeter stations of
the SoilCan network can be found in Pütz et al. (2016). Three lysimeters were moved from
Rollesbroich to Selhausen in November 2011. In this study, we make use of measurements
made in a six-year period from 2013 to 2018.
Table 2 summarizes the annual average water balances measured in the six lysimeters in the
six-year period from 2013 to 2018, as well as the average annual harvested biomass and
calculations of the water use efficiency, defined as the ratio of harvest to evapotranspiration.
In the wet climate at Rollesbroich, actual evapotranspiration was ca. 90% of the potential rate
calculated by the FAO version of the Penman-Monteith equation for the period 2013-2018
(641 and 710 mm/year respectively), while percolation from the lysimeters was on average
42% of the precipitation (442 and 1062 mm/year respectively). Thus, evapotranspiration at
Rollesbroich is mostly limited by the available energy and is only rarely limited by water supply
(Gebler et al., 2015; Rahmati et al., 2020). Notably, the ratio of actual to potential
evapotranspiration was only slightly smaller in the much drier climate of Selhausen than at
Rollesbroich (on average 86%, Table 2). Figure 1 shows that a strong limitation of the water
supply on evapotranspiration at Selhausen can only be seen in the very dry year of 2018, when
the ratio between actual and potential rates fell to ca. 60%. It is also striking that the actual
evapotranspiration slightly exceeds precipitation at Selhausen, so that the net percolation at
the base of the lysimeters is negative (i.e. an upwards directed flow; Table 2).
Table 2 shows that the differences in water balance components among the three replicate
lysimeters at both sites are very small. For precipitation, the difference between the largest
and smallest measured totals among the replicates at Rollesbroich and Selhausen is only ca.
3% and 1% of the mean value respectively. Furthermore, the difference in evapotranspiration
between the two lysimeters with the largest and smallest values is equivalent to only 1% of
the precipitation at Selhausen and 2.6% of the precipitation at Rollesbroich. This limited
within-site variation in hydrologic response appears to be consistent with the available data
for soil water contents and pressure heads. The 'in situ' water retention data (Figure S3 and





Table S1) suggest that there is limited spatial variation in soil hydraulic properties among the
six lysimeters. Percolation is somewhat more variable (Table 2), despite the fact that the
pressure heads in the surrounding soil at 1.4 m depth controlling water flow at the base of the
lysimeter are also quite similar among the replicates, especially at Rollesbroich (see figure S4).
Likewise, harvested biomass at Selhausen was similar in all three replicate lysimeters, whereas
it varied more at Rollesbroich, with one lysimeter clearly an outlier (Ro_Y_013, Table 2). Much
larger nitrate nitrogen concentrations were consistently found at the beginning of the
experiment in the leachate from this lysimeter (Giraud et al. 2021), which suggests that the
larger harvest from Ro_Y_013 may be due to a better nutrient supply from the soil. Table 2
and figure 2 show that the water use efficiency (WUE) of the grassland in the drier climate at
Selhausen was smaller than for the lysimeters at Rollesbroich (Forstner et al., 2021), since
harvests were somewhat smaller and evapotranspiration was larger.
In the following, we assess the capability of a relatively simple (parsimonious) eco-hydrological
model to match the data measured in the replicate lysimeters in the two contrasting climates
at Rollesbroich and Selhausen. We also use the model to identify plausible reasons for the
differences in soil hydrology and grassland growth observed between the sites.
**2.2 Model description**
2.2.1 Potential evapotranspiration
In the longer term, the extent of grass cover can be affected by a changing climate, which will
alter the energy balance partitioning at the land surface. We therefore employ the dual-source
Penman-Monteith equation (Shuttleworth and Wallace, 1985; Shuttleworth and Gurney,
1990), which enables the estimation of potential soil evaporation $E_p$ (m day$^{-1}$) and potential
transpiration $T_p$ (m day$^{-1}$) from dynamic plant properties and meteorological variables:
$$E_p = \frac{C_s}{\lambda} \left[ \frac{\Delta R_n + \left\{ \frac{\rho\, c_p\, VPD - \Delta\, r_a^s \left( R_n - R_{n(s)} \right)}{r_a^a + r_a^s} \right\}}{\Delta + \gamma \left( 1 + \left( \frac{r_s^s}{r_a^a + r_a^c} \right) \right)} \right] \qquad (1)$$
$$T_p = \frac{C_c}{\lambda} \left\{ \frac{\Delta R_n + \left\{ \frac{\rho\, c_p\, VPD - \Delta\, r_a^c\, R_{n(s)}}{r_a^a + r_a^c} \right\}}{\Delta + \gamma \left( 1 + \left( \frac{r_s^c}{r_a^a + r_a^c} \right) \right)} \right\} \qquad (2)$$
$$C_s = \frac{1}{1 + \left( \frac{R_S R_a}{R_C (R_S + R_a)} \right)} \qquad (3)$$
$$C_c = \frac{1}{1 + \left( \frac{R_C R_a}{R_S (R_C + R_a)} \right)} \qquad (4)$$
$$R_a = (\Delta + \gamma) r_a^a \qquad (5)$$
$$R_c = (\Delta + \gamma) r_a^c + \gamma r_s^c \qquad (6)$$
$$R_s = (\Delta + \gamma) r_a^s + \gamma r_s^s \qquad (7)$$





where λ is the latent heat of vapourisation, ρ is the air density, $C_p$ is the specific heat of air,
VPD is the vapour pressure deficit, Δ is the slope of the saturation vapour pressure curve, γ is
the psychrometer constant, $r_s^s$ is the surface resistance of wet soil (here fixed at 20 s m⁻¹), $r_s^c$
and $r_a^c$ are the bulk unstressed stomatal and boundary layer resistances of the canopy, $R_n$ and
$R_{n(s)}$ are the net radiation above and below the canopy and $r_a^s$ and $r_a^a$ are the aerodynamic
resistances from soil to canopy and canopy to the reference height (= 2m) respectively, both
of which are estimated from wind speed and crop height following the approach described by
Shuttleworth and Gurney (1990) and Zhou et al. (2006). Assuming that only half the leaf area
contributes to transpiration, the canopy surface resistance $r_s^c$ (s m⁻¹) can be expressed as:
$$r_s^c = \frac{2}{\{k_{sto(max)} f_L f_{t(c)}\} LAI} \tag{8}$$
where $k_{sto(max)}$ is the maximum leaf stomatal conductance (m s⁻¹), $LAI$ is the leaf area index (m²
m⁻²), $f_{t(c)}$ is a function describing the response of conductance to air temperature (see
*Environmental stress functions*) and $f_L$ is a light response function given by:
$$f_L = \left(\frac{R_i}{R_i + R_{50}}\right) \tag{9}$$
where $R_i$ is the incoming radiation (MJ m⁻² d⁻¹) and $R_{50}$ is the half-saturation constant for light
(here fixed at 5 MJ m⁻² d⁻¹). The bulk boundary layer resistance $r_a^c$ (m s⁻¹) is given by:
$$r_a^c = \frac{r_b}{LAI} \tag{10}$$
where $r_b$ is the leaf boundary layer resistance (here fixed at 25 s m⁻¹). Radiation interception
by the plant canopy is calculated using Beer's law:
$$R_{n(s)} = R_n(1 - f_{int}) \tag{11}$$
$$f_{int} = 1 - e^{-\beta LAI} \tag{12}$$
where $f_{int}$ is the fraction of the net radiation intercepted by the plant canopy and $\beta$ is the
extinction coefficient. Net radiation is estimated from incoming solar radiation $R_i$ using the
algorithms described in Allen et al. (1998).
Rainfall interception is at present not considered in the model. Although interception losses
may not be negligible even for a reasonably short grassland plant community (Ataroff and
Naranjo, 2009; Hu et al., 2009; Groh et al., 2019), we assume that the errors introduced by
ignoring the net increase in evaporation due to rainfall interception will be negligible.
2.2.2 Water flow, root water uptake and transpiration
Some SVAT models use tipping bucket or reservoir models to describe water storage and flow
in the soil, even though physical approaches based on Richard's equation are not difficult to
parameterize and usually perform better (e.g. Diekkrüger et al., 1995; Kröbel et al., 2010;
Guest et al., 2017). Water uptake by plant roots is also represented empirically in many widely
used SVAT models (Wang and Smith, 2004; Smithwick et al., 2014). These two issues are to
some extent linked, as physics-based models of root water uptake require information on soil
water pressures, while tipping bucket or reservoir models only simulate soil water contents.



In principle, water uptake by roots also depends on the 3D architecture of the plant root
system as well as the hydraulic properties along multiple flow pathways in the soil and plant
(e.g. Raats, 2007). Physics-based models have been developed that can calculate water flow
and uptake by a root system explicitly defined in 3D (e.g. Dunbabin et al., 2013; Schnepf et al.,
2018). Although some attempts have been made (e.g. Postma et al., 2017; Mboh et al., 2019),
these models are not so well suited to coupling to SVAT models due to their high parameter
and computational requirements. However, some parsimonious physics-based macroscopic
approaches have been developed (e.g. de Jong van Lier et al., 2008, 2013; Couvreur et al.,
2012; Javaux et al., 2013; Sulis et al., 2019) that contain no more parameters than the
empirical models. The parameters of these models are also easier to estimate since they have
a stronger physical basis (de Willigen et al., 2012; Javaux et al., 2013). For the same reason,
the predictive use of these models should also be more robust in principle. The simplest
physics-based models (e.g. Raats, 2007; de Jong van Lier et al., 2008) only describe flow to the
roots and neglect flow and resistances within the plant. In this study, we use the model of root
water uptake described by de Jong van Lier (2008), which is coupled with Richards' equation
to calculate transient water flow soil water content, $\theta$ (m m$^{-3}$) in a one-dimensional soil profile:
$$\frac{d\theta}{dt} = \frac{d}{dz}\left[K(\theta)\left(\frac{d(\psi+z)}{dz}\right)\right] - U \tag{13}$$
where $t$ is time (days), $z$ is height (m), $K$ is the soil hydraulic conductivity (m day$^{-1}$), $\psi$ is the
pressure head (m) and $U$ (days$^{-1}$) is the so-called sink term which accounts for root water
uptake. The bottom boundary condition required to solve Richards' equation is specified as
the known (measured) pressure head at the base of the simulated soil profile, i.e. at 1.4 m
depth. The upper boundary condition to equation 13 is specified as a flux given by the
difference between the known precipitation rate and the actual soil evaporation, $E_a$, which in
turn is given by:
$$E_a = \min(q_{max}; E_p) \tag{14}$$
where $q_{max}$ is the maximum flow rate towards the soil surface calculated using Darcy's law
from the pressure head in the uppermost soil layer. The soil water retention and hydraulic
conductivity functions required to solve equation 13 are given by the Mualem-van Genuchten
model (Mualem, 1976; van Genuchten, 1980), with the matching point hydraulic conductivity,
$K_{10}$ (m day$^{-1}$) set at a pressure head of -0.1 m (Luckner et al., 1989) and assuming that the
residual water content is negligible:
$$S = \frac{\theta}{\theta_s} \tag{15}$$
$$S = (1 + |\alpha\,\psi|^n)^{\frac{1}{n}-1} \tag{16}$$
$$K(S) = K_{10}\left(\frac{S}{S_{10}}\right)^{\tau}\left[\frac{1-\left(1-S^{\left(\frac{n}{n-1}\right)}\right)^{\left(1-\frac{1}{n}\right)}}{1-\left(1-S_{10}^{\left(\frac{n}{n-1}\right)}\right)^{\left(1-\frac{1}{n}\right)}}\right]^2 \tag{17}$$





where $S$ is the degree of saturation (-), $S_{10}$ is the value of $S$ at a pressure head of -0.1 m, $\theta_s$ is
the saturated water content (m³ m⁻³), $\alpha$ (m⁻¹) and $n$ (-) are shape parameters and $\tau$ is a
tortuosity/connectivity factor.
Neglecting water storage changes in the plants, the total water uptake from the root zone
equals the actual transpiration rate, $T_a$, such that:
$T_a = \sum_i U_i \Delta z_i$         (18)
where the subscript $i$ refers to a layer in the root zone and $\Delta z$ is its thickness. To calculate the
sink term $U_i$ and actual transpiration $T_a$, we make use of the parsimonious physics-based
model of root water uptake proposed by de Jong van Lier et al. (2008), which implicitly
accounts for compensatory uptake (Jarvis, 2011). Neglecting plant resistances, they derived
the macroscopic water uptake sink term to Richards' equation by upscaling a model of water
flow to a single root based on the concept of matric flux potential $M$ (m² day⁻¹):
$M_i = \int_{\psi_w}^{\psi} K(\psi)d\psi$         (19)
where $\psi_w$ is the soil water pressure head at which water uptake by plants ceases. At the
microscopic scale in the soil, $M$ will continuously decrease towards its value at the root/soil
interface $M_o$. In this study, we used the approximate solution derived by de Jong van Lier et
al. (2009) to calculate $M$ for the van Genuchten-Mualem model of soil hydraulic properties.
Assuming that $M_o$ is constant in the root zone and neglecting the effects of root and plant
resistances on flow through the soil-plant system, de Jong van Lier et al. (2008) showed that
the sink term for water uptake by roots in each soil layer can be expressed as:
$U_i = \rho_i(M_i - M_0)$         (20)
where $\rho$ is a composite root parameter (m⁻²) given by (de Jong van Lier, 2008):
$\rho_i = \dfrac{4}{r_o^2 - a^2 r_{m(i)}^2 + 2\left(r_o^2 + r_{m(i)}^2\right)LN\left(\frac{a\,r_{m(i)}^2}{r_o^2}\right)}$         (21)
where $r_o$ is the root radius, $a$ is the distance to the root (normalized by $r_m$) at which the soil
water content is equal to the average value in layer $i$ (fixed here at 0.53; de Jong van Lier et
al., 2008) and $r_m$ is the mean half distance to the root surface, which can be calculated from
the effective root length density $R_{LD(i)}$ (m m⁻²) as:
$r_{m(i)} = \sqrt{\dfrac{1}{\pi R_{LD(i)}}}$         (22)
Actual transpiration is determined by the minimum of the potential transpiration rate, $T_p$, and
the maximum possible flow rate of water to the root system, $T_{max}$, which occurs when $M_o$=0
(see equations 18 and 20). Thus, actual transpiration can also be expressed as:
$T_a = \min(T_{max}; T_p)$         (23)
where $T_{max}$ is obtained by combining equations 18 and 20 with $M_o$=0:
$T_{max} = \sum_i \rho_i M_i \Delta z_i$         (24)



For unstressed plants, $T_{max} \geq T_p$ and $T_a = T_p$. In this case, the unknown value of $M_o$ in equation
20 is calculated by combining equations 18, 20 and 24 and knowing that $T_a = T_p$, which gives:
$$M_0 = \frac{T_{max} - T_p}{(\sum_i \rho_i \Delta z_i)} \qquad ; \qquad T_{max} \geq T_p \qquad\qquad (25)$$
$$M_0 = 0 \qquad ; \qquad T_{max} < T_p$$
It can be seen from equations 24 and 25 that in any given soil, plant water stress will set in
earlier when potential transpiration rates are high and total root length density is low.
2.2.3 Growth model for perennial grassland
Even though detailed growth models designed for perennial forage grass are already available
(e.g. Schapendonk et al., 1998; Jing et al., 2012; Persson et al., 2014; Kellner et al., 2017), we
developed a simple generic model for the purpose of this study, which only simulates
vegetative growth. This model is intended to be able to capture the main longer-term
feedback mechanisms between soil water status and grass growth (Tardieu and Parent, 2017)
and is designed to be compatible with simpler water uptake models that do not simulate water
potentials, resistances and flows within plants (Manzoni et al., 2013).
In the model, net assimilation is calculated using the concept of radiation use efficiency (e.g.
Sinclair and Muchow, 1999), which implicitly assumes a constant ratio of respiration to
photosynthesis (i.e. carbon use efficiency; Gifford, 2003). Furthermore, we assume that
assimilation is limited by light, water and temperature, but not by sub-optimal nutrition. The
allocation of assimilates to above- and below-ground biomass depends on environmental
stressors. In this respect, based on empirical knowledge, we assume that water stress and sub-
optimal temperatures will increase the partitioning of assimilates to roots (e.g. Jones et al.,
1980a; Kahmen et al., 2005; Hui and Jackson, 2006; Wedderburn et al., 2010; Skinner and
Comas, 2010; Padilla et al., 2013; Nosalewicz et al., 2018; Meurer et al., 2019). Excess
carbohydrates produced by grasses during periods of "sink-limited" growth are stored as non-
structural reserves, mostly in the tiller bases and roots (Thomas, 1991; Johansson, 1993;
Volaire et al., 1998; Thomas and James, 1999; Østrem et al., 2011; Martínez-Vilalta et al., 2016;
Hofer et al., 2017; Katata et al., 2020). These non-structural carbohydrates contribute to rapid
recovery of growth after drought or defoliation by grazing or harvesting (Morvan-Bertrand et
al., 1999; Jing et al, 2012; Schmitt et al., 2013; Benot et al., 2019). However, for the sake of
simplicity, our growth model only tracks total biomasses in above- and below-ground
compartments and does not explicitly account for reserves of non-structural carbohydrates.
The loss of both above- and below-ground biomass by diverse mechanisms (e.g. herbivory,
exudation, root decay) is modelled in a simple way as a lumped first-order process. Although
root longevity can be affected by drought (e.g. Chen and Brassard, 2013), this is neglected in
the model for reasons of simplicity. Root systems also show plastic responses to
environmental conditions, such that growth of new roots takes place where water is easily
available, while root dieback occurs in dry soil (e.g. Jupp and Newman, 1987; DaCosta et al.,
2004; Wedderburn et al., 2010). Dynamic modeling of root proliferation and loss in response
to soil conditions remains a very difficult task (e.g. Wang and Smith, 2004; Boote et al., 2013;





Smithwick et al., 2014; Stöckle and Kemanian, 2020). Here, for the sake of simplicity, we
assume that the distribution of root biomass and length within the root zone are constant, as
well as the maximum depth of roots in the profile. With these assumptions, changes in the
below-ground (root) biomass in any soil layer $i$, $B_{bg(i)}$ (kg dry matter m$^{-2}$) are given by:
$$\frac{dB_{bg(i)}}{dt} = f_{bg}A\,f_{r(i)} - k_{bg}B_{bg(i)} \qquad\qquad (26)$$
where $k_{bg}$ is a first-order rate constant for root biomass loss (d$^{-1}$), $A$ (kg m$^{-2}$ d$^{-1}$) is the dry matter
assimilation rate, $f_{bg}$ is the fraction of dry matter production partitioned to roots and $f_{r(i)}$ is the
fraction of this root production allocated to layer $i$, which is prescribed by a logistic dose
response function (Schenk and Jackson, 2002; Fan et al., 2016; Metselaar et al., 2019):
$$f_{r(i)} = \left[\frac{1}{1+\left(\frac{D_U}{D_{50}}\right)^c}\right] - \left[\frac{1}{1+\left(\frac{min(D_L;D_r)}{D_{50}}\right)^c}\right] \quad ; \qquad D_r > D_U \qquad (27)$$
$$f_{r(i)} = 0 \qquad\qquad\qquad\qquad\qquad ; \qquad D_r \le D_U$$
where $c$ is a shape factor, $D_U$ and $D_L$ are the depths to the upper and lower boundaries of layer
$i$, $D_r$ is an effective root depth, which we define as the depth above which 95% of the roots are
located and $D_{50}$ is the depth above which 50% of the root biomass is found, such that:
$$D_{50} = \frac{D_r}{\left(\frac{1}{0.95}-1\right)^{\frac{1}{c}}} \qquad\qquad (28)$$
With this approach, 5% of the roots are located below the maximum root depth. In the model,
we distribute this extra root biomass to the uppermost two numerical layers in equal amounts.
The assimilation rate $A$ in equation 26 is calculated as a function of incoming solar radiation
$R_s$ (MJ m$^{-2}$ day$^{-1}$) and two dimensionless stress functions, $f_{t(p)}$ and $f_{w(p)}$ varying between zero
and unity to represent the effects of temperature and water stress on dry matter production:
$$A = f_{int}\,R_s\,RUE_{max}f_{t(p)}f_{w(p)} \qquad\qquad (29)$$
where $RUE_{max}$ is the maximum radiation use efficiency (kg MJ$^{-1}$). The root allocation fraction
$f_{bg}$ in equation 26 is calculated as a function of plant stressors (i.e. air temperature, water
stress) and "sink strength", represented here by the fraction of radiation intercepted, $f_{int}$, using
an approach based on the simple model concept outlined by Friedlingstein et al. (1999):
$$f_{bg} = f_{bg(opt)}\left(\frac{2\,f_{int}}{f_{int}+min(f_{t(a)}\,;\,f_{w(a)})}\right) \qquad\qquad (30)$$
where $f_{bg(opt)}$ is the fraction of assimilates partitioned below-ground when the conditions for
above-ground production are optimal (i.e. full canopy, optimal temperature and no water
stress) and $f_{t(a)}$ and $f_{w(a)}$ are response functions to account for the effects of sub-optimal
conditions of temperature and water on allocation. With this approach, sub-optimal
environmental conditions (extreme air temperatures, plant water stress) increase the
proportion of assimilates partitioned to roots, whereas a loss of leaf area (e.g. due to harvest)
triggers an increased allocation of assimilates to the above-ground biomass (see figure S5).





Changes in above-ground biomass, $B_{ag}$ (kg m$^{-2}$) are given by:
$$\frac{dB_{ag}}{dt} = (1 - f_{bg})A - k_{ag}max(1 - f_{t(a)}; 1 - f_{w(a)})B_{ag} - \Gamma\left(1 - \frac{H_{cut}}{H}\right)\left(\frac{B_{ag}}{\Delta t}\right) \quad (31)$$
where $\Gamma$ is a binary variable, indicating the occurrence of harvest of above-ground biomass
(zero for no harvest, 1 for harvest), $H_{cut}$ is the cutting height at harvest (here set to 0.01 m), $H$
is the grass height at harvest (m), $\Delta t$ is the time step in the model and $k_{ag}$ is a rate coefficient
(d$^{-1}$) regulating the loss of above-ground biomass by senescence and leaf fall, which is also
promoted by sub-optimal temperatures or plant water stress, employing the same empirical
functions used for assimilate partitioning between above-and below-ground biomass. In this
model, we do not account for standing dead above-ground biomass, which would alter the
partitioning of solar radiation between soil and plant, without contributing to transpiration
and assimilation, since we assume that the loss of green leaf area results in immediate litter-
fall. However, it would be straightforward to incorporate standing dead biomass in future
versions of the model, for example in the way described by Montaldo et al. (2005).
Feedbacks from the plant growth model to the hydrological model are provided by the leaf
area index, LAI, and effective root length density, $R_{LD(i)}$, which are calculated as:
$$LAI = B_{ag}S_{leaf} \quad (32)$$
$$R_{LD(i)} = \varepsilon\left(\frac{B_{bg(i)}}{z_i}\right)S_{root} \quad (33)$$
where $S_{leaf}$ (m$^2$ kg$^{-1}$) and $S_{root}$ (m kg$^{-1}$) are the specific leaf area and specific root length and $\varepsilon$ is
the fraction of the total root length that is effective for water uptake (Faria et al., 2010). The
height of the crop also acts as a feedback control on the water balance, since it affects the
aerodynamic resistances to evapotranspiration (equations 1 to 7). The height of the grass
cover is not explicitly simulated in our relatively simple growth model. Instead, we calculate
plant height as a function of simulated LAI, based on the data from both sites (see figure S6).
2.2.4 Environmental stress functions
As in other models of crop growth (Wu et al., 2016), we use the ratio of actual to potential
transpiration to represent the effects of water stress on assimilation via stomatal closure:
$$f_{w(p)} = \frac{T_a}{T_p} \quad (34)$$
Water stress also limits crop growth without affecting photosynthesis by several different
mechanisms (Körner, 2015; White et al., 2016; Tardieu et al., 2018; Loka et al., 2019; Gupta et
al., 2020). Many crop models calculate limitations on leaf growth as a threshold function of
the soil water deficit in the root zone. Here, we make use of the matric flux potential at the
root surface $M_o$ (see equations 20 and 25) as a measure of plant water stress, since it should
be more physically and physiologically meaningful. We therefore define a second water stress
index as a threshold response function of $M_o$, varying between zero and unity, which regulates
dry matter allocation and leaf loss in the model (equations 30 and 31):



$\quad f_{w(a)} = 1 \qquad ; \qquad M_o \geq M_{o(crit)}$ (35)
$\quad f_{w(a)} = \dfrac{M_o}{M_{o(crit)}} \qquad ; \qquad M_o < M_{o(crit)}$
where $M_{o(crit)}$ is a critical value of $M_o$, which is in turn calculated from a user-defined value of
a critical pressure head at the soil/root interface, $\psi_{o(crit)}$.
As in many soil-crop models (Wu et al., 2016), the temperature response function in equations
8 and 29 to 31 is modelled with a piece-wise linear function (figure S7):
$\quad f_{t(c,p,a)} = 0 \qquad\qquad\qquad\quad ; \qquad T < T_b \; or \; T > T_c$ (36)
$\quad f_{t(c,p,a)} = \left( \dfrac{T-T_b}{T_{o(low)}-T_b} \right) \qquad ; \qquad T_b \leq T \leq T_{o(low)}$
$\quad f_{t(c,p,a)} = \left( \dfrac{T_c-T}{T_c-T_{o(high)}} \right) \qquad ; \qquad T_{o(high)} \leq T \leq T_c$
$\quad f_{t(c,p,a)} = 1 \qquad\qquad\qquad\quad ; \qquad T \geq T_{o(low)} \; and \; T \leq T_{o(high)}$
where $T$ is the mean air temperature (°C), $T_{o(low)}$ and $T_{o(high)}$ define the optimum temperature
(°C) range at which $f_{t(p,a)}$ equals unity and $T_b$ and $T_c$ are the base and ceiling temperatures (°C)
at which the function equals zero. Different values for the parameters in equation 36 can be
assigned for transpiration ($f_{t(c)}$), assimilation ($f_{t(p)}$) and allocation and leaf fall ($f_{t(a)}$).
**2.3 Model application**
2.3.1 Modelling strategy
In this study, uncertainty in the model parameterization has been addressed through Monte
Carlo simulations following the GLUE methodology (see *Sensitivity and uncertainty analysis*).
In principle, it would be possible to apply the model individually to each lysimeter in such an
approach. However, this would have been far too demanding of computer resources. Instead,
recognizing the comparatively small differences in hydrological behavior among the three
replicates at each site (Table 1) and the fact that the same soil type is present at both sites,
we decided to simplify the analysis by assuming a common parameterization for the soil
hydraulic properties in all six lysimeters. Similarly, we also neglected the small differences in
boundary conditions among the replicate lysimeters at each site. Thus, precipitation (Table 1;
figure S1) and pressure heads at the bottom boundary (figure S4) measured for one lysimeter
at each site (Ro_Y_015 at Rollesbroich and Se_Y_026 at Selhausen) were used to represent all
three replicates. This approach also implicitly assumes that we can neglect the likelihood of
small differences in initial conditions among the replicates at each site. Initial soil water
pressure head profiles at each site were set according to the results of preliminary simulations
involving "trial and error" calibration to measured early time water outflows from the
lysimeters. Initial above- and below-ground plant biomasses were calculated assuming that
the roots constituted 80% of the total biomass and that the initial leaf area index was 1.5. It
can be noted that model predictions quickly become independent of these initial guesses.


2.3.2 Soil hydraulic parameters
Four horizons were identified from a soil profile description at the Rollesbroich site (Table 1).
Common parameters of the Mualem-van Genuchten model were estimated for each horizon
from a combination of direct measurements and pedotransfer functions (Table 3). The paired
TDR and tensiometer measurements obtained in the lysimeters at 30 and 50 cm depth were
utilized to estimate common water retention parameters at the two sites for the horizons at
24-48 and 48-90 cm depth by least-squares fitting (Table 3 and figure S3). We used the HYPRES
class pedotransfer functions (Wösten et al., 1999) to estimate the van Genuchten water
retention parameters from the soil textural class in the deep subsoil (90-140 cm depth) where
no data was available. The measurements from the matric potential sensors installed in the
uppermost soil horizon (0-24 cm depth) appeared to be unreliable. We therefore also used
the HYPRES pedotransfer functions to estimate the shape parameter $n$ in the topsoil, while $\alpha$
was set equal to the same value as the deeper horizons. Saturated water contents clearly
differed between the two sites in the uppermost horizon and were estimated from the data
by eye. Hydraulic conductivity at a pressure head of -10 cm (see table 3) was estimated from
clay content in each horizon using the pedotransfer function developed by Jarvis et al. (2013).
2.3.3 Sensitivity and uncertainty analysis
A comprehensive uncertainty analysis treating a large number of model parameters as
uncertain was not feasible in this study from the point of view of both data support and
computational capacity, even for the comparatively parsimonious model used in this study.
We therefore performed a preliminary Monte Carlo sensitivity analysis to support the
selection of a limited number of parameters to include in the uncertainty analysis. We ran 500
simulations for each site for the period 2013-2018 with parameter values obtained by Latin
hypercube sampling from uniform distributions (table S2 in the supplementary information).
We quantified the sensitivity of two target outputs (i.e. total evapotranspiration and harvest
during the experimental period) to model parameters using Spearman rank partial correlation
coefficients. The sampled ranges for the plant parameters in the model were selected to
reflect variations based on information in the literature. Three soil hydraulic parameters were
also included in this analysis ($K_{10}$, $\alpha$ and $n$). This was done by applying scaling factors (see table
S2) to the parameter values in Table 3 to broadly reflect the uncertainty arising from the use
of pedotransfer functions as well as the spatial variations in the water retention curves derived
from the lysimeter measurements (figure S3). It should be noted here that the resulting ranges
adopted for the two van Genuchten parameters encompass the differences found among the
six lysimeters at both depths. Table S2 shows the results. In general, evapotranspiration and
harvest is much more sensitive to many of the plant parameters than to variation in the soil
hydraulic properties, which lends support to a modelling strategy in which soil hydraulic
properties are set to identical values for all lysimeters. We therefore focused the uncertainty
analysis on investigating differences in key plant parameters between the two sites.
Of the many highly sensitive plant parameters (Table S2), we decided to treat four as
uncertain: the radiation extinction coefficient $\beta$, the maximum stomatal conductance $k_{sto(max)}$,
the maximum root depth $D_r$ and the limiting pressure head $\psi_{o(crit)}$ that controls dry matter
(DM) allocation between above- and below-ground compartments as well as the rate of leaf



loss. Several subjective criteria underpin this selection. Firstly, they are among the most highly
sensitive parameters for both evapotranspiration and harvest yields (Table S2). In this respect,
with the exception of $T_{o(low)}$, it seems that plant parameters controlling temperature response
are much less sensitive than those regulating water stress (Table S2). Secondly, in addition to
the changes in plant community composition, there are also some known mechanisms of plant
acclimation (e.g. Vincent et al., 2020) that could explain why these four parameters might
plausibly take different values at the two sites. Finally, the effects on these four model
parameters on the model outputs are unlikely to be strongly correlated with one another. This
would not be the case for some of the other sensitive parameters. For example, the radiation
extinction coefficient $\beta$ would be correlated with the maximum radiation use efficiency, while
$\psi_{o(crit)}$ would be correlated with both the parameter controlling DM allocation under optimal
conditions, $f_{bg(opt)}$, as well as the effective root fraction, $\varepsilon$. The remaining plant parameters in
the model were therefore set to fixed values estimated from data in the literature (Table 4),
prioritizing field studies rather than pot experiments, as the development of drought and the
plant response to stress are known to be strongly affected by restricted root zones (Jones et
al., 1980a,b). Specific leaf area was set to 142 $cm^2$ $g^{-1}$ based on the measurements of above-
ground biomass and leaf area index for the combined dataset at both sites (see figure S6). The
relationship shown in figure S6 shows some scatter, but no systematic difference between the
sites is apparent. In this respect, Norris (1982) also found no significant differences in specific
leaf area for *Lolium perenne* in droughted, control and irrigated plots.
We used the GLUE (Generalized Likelihood Uncertainty Estimation; Beven and Binley, 1992;
Beven 2006) methodology to account for parameter uncertainty. The objective of this
informal Bayesian approach is not to find a single optimum parameter set by calibration, as it
acknowledges that many different parameterizations will perform equally well (so-called
"equifinality"), not least as a consequence of the inevitability of model (structural) error. The
objective of GLUE is therefore to identify acceptable ("behavioural") parameterizations. To
support this analysis, we ran 2000 simulations for each site, with parameter sets determined
using Latin Hypercube sampling from the prior uncertainty ranges for the four uncertain
parameters shown in Table 5. GLUE involves several subjective decisions, two of the most
important ones being the choice of a likelihood function (i.e. a measure of goodness-of-fit)
and deciding on the criteria that should be used to determine whether a simulation is
acceptable or not. We considered that a parameterization was acceptable if two criteria were
satisfied. The first uses calculations of the model efficiency, *ME*, for the six observed time
series of data (i.e. water contents at three depths, evapotranspiration rates, LAI, harvests):
$$ME = \frac{\sum_{i=1}^{m}(O_i - \bar{O})^2 - \sum_{i=1}^{m}(O_i - P_i)^2}{\sum_{i=1}^{m}(O_i - \bar{O})^2} \qquad (37)$$
where *O* and *P* are the observed and simulated values for a given data type and *m* is the
number of observations. The maximum value of *ME* is one, when predictions and observations
are identical, while a negative value implies a poor model, since it means that taking the
average of the observations would give a better prediction. A simulation was considered
acceptable if i.) the model efficiency for all six data types was within 0.5 of the maximum value
for that data series, and ii.) both the simulated annual average evapotranspiration *AET*



(mm/year) and overall (apparent) water use efficiency *WUE* (kg DM m⁻³) were within
acceptable limits roughly defined by the observations (see Table 2):

565        *At Rollesbroich*: $610 < AET < 660 \ and \ 1.0 < WUE < 1.2$

566        *At Selhausen*: $680 < AET < 730 \ and \ 0.85 < WUE < 1.05$

This second criterion ensures that the acceptable parameterizations respect the overall broad
differences observed in the water balance components and harvest yields between the two
sites. Note that the acceptable limit for WUE at Rollesbroich makes no attempt to "honour"
the data from lysimeter Ro_Y_013, since it is considered an outlier, as discussed earlier. In
total, 35 simulations at Rollesbroich and 57 at Selhausen satisfied these criteria. It is desirable
to have the same number of acceptable parameter sets at each site. From these acceptable
simulations, we therefore selected the 30 best simulations at each site (i.e. 1.5% of the total
number of simulations) according to the average model efficiency for the six data types.
## 3. Results and discussion
### 3.1 Acceptable parameter values
The distributions of the acceptable values for the four uncertain parameters are shown in
figure 3, while posterior parameter ranges defined by different percentiles of these
distributions are presented in table 5. The posterior uncertainty ranges are much smaller than
the prior uncertainty ranges, which suggests that values for all four uncertain parameters
were clearly identifiable from the data. No differences between the two sites were found for
two of the parameters, the radiation extinction coefficient β and $\psi_{o(crit)}$ the parameter
controlling dry matter allocation and leaf loss as a function of water stress ($p$ = 0.98 and 0.16
respectively). The derived values of $\psi_{o(crit)}$ (median value of -271 cm at both sites, Table 5) are
much larger than $\psi_w$ (= -150 m, Table 4), which indicates that water stress affects above-
ground plant growth long before stomatal closure limits transpiration and assimilation
(Staniak and Kocoń 2015; Körner, 2015; Loka et al., 2019). This has been shown experimentally
for droughted field-grown grass/clover pastures by Jones et al. (1980a,b) and Hofer et al.
(2017). The values of the radiation extinction coefficient (inter-quartile range = 0.51-0.65 at
both sites) are typical of values reported for grassland ecosystems (Zhang et al., 2014).
In contrast, the results of the GLUE analysis suggest that both the maximum root depth and
the unstressed stomatal conductance have increased significantly for the lysimeters moved to
Selhausen ($p < 0.0001$ for both). The estimated root depth at Rollesbroich (ca. 56 cm) matches
observations made at the site at the time of extraction of the lysimeters reasonably well. The
simulations suggest that the maximum root depth at Selhausen has increased to ca. 80 cm,
while the maximum stomatal conductance has roughly doubled. The mechanisms underlying
these changes are not clear. One reason may be the significant changes observed in the plant
community composition at Selhausen compared with the original resident plant community
(figure S2), as plant traits may differ significantly between herbs and grasses. Another likely
reason is that one or more of the dominant species adapted to the new climate. In this respect,
plants are known to acclimatize to environmental stresses at a range of time-scales by various
physiological and morphological mechanisms (e.g. Maseda and Fernández, 2006; Nicotra et



al., 2010; Nicotra and Davidson, 2010; Manzoni et al., 2013; Bartlett et al., 2014; Tardieu et
al., 2018; Vincent et al., 2020).
**3.2 Soil hydrology**
Figures 4 and 5 show comparisons of the acceptable simulations at the two sites with the soil
water contents measured at the three depths in the lysimeters and daily evapotranspiration
rates respectively. The model efficiencies for these simulations are shown in table 6. Figure 6
compares measured annual average evapotranspiration and percolation in the period 2013-
2018 with the simulations. Taken together, these results show that the model performs very
well, matching the temporal dynamics in the high-time resolution data on state variables and
fluxes as well as reproducing the differences in the overall water balances at the two sites.
This is probably because the macroscopic sink term describing root water uptake that we
coupled to Richards' equation has a reasonably strong physical basis. In particular, this model
implicitly accounts for the mechanism of "compensatory" root water uptake, something which
is clearly necessary in order to reproduce the extensive drying in the root zone observed in
the Selhausen lysimeters, with very little reduction in water uptake and transpiration.
Figure 7 shows some terms of the simulated water balances that were not measured. Potential
evapotranspiration calculated internally in the model by the Shuttleworth-Wallace version of
the Penman-Monteith equation as a dynamic function of leaf area development at the two
sites is very similar to the estimates obtained by the FAO version (Figure 7; table 2), which
only treats the vegetation implicitly. This is in spite of the fact that the balance between
simulated soil evaporation and transpiration differs strongly between the two sites, with soil
evaporation being a much larger component of the water balance at Rollesbroich (Figure 7),
where it comprises ca. 70% of the total evapotranspiration. There may be several reasons why
soil evaporation is such an important term in the water balance at Rollesbroich, including the
wet climate with high wind speeds (Groh et al., 2019) the capillary nature of the soil and also
the fact that the grassland is harvested 3-4 times during the growing season, which exposes
the soil surface to evaporation. In contrast, soil evaporation is much smaller (ca. 50% of total
evapotranspiration) in the drier climate at Selhausen, presumably because drying of the soil
surface in summer frequently reduced evaporation below the potential rate (figure 7).
Figure 7 shows that the model simulates only small reductions of transpiration due to water
stress and stomatal closure at both sites ($T_a < T_p$), which matches the inference derived from
comparing the lysimeter data with the FAO estimates of potential evaporation (figure 1). This
result is not especially surprising for the grassland growing in the wet climate at Rollesbroich,
but it does require further analysis and explanation for the much drier Selhausen site. Figure
8 shows the simulated time-courses of the two water stress functions in the model. Short
periods of stomatal closure induced by water stress occur every summer at Selhausen in most
of the acceptable model simulations, with one more extended period of drought stress (ca. 1
to 2 weeks) in 2018. However, overall, the extent and severity of reductions in transpiration
due to water stress simulated at Selhausen is not much larger than at Rollesbroich. The reason
for this becomes apparent from a comparison of the results for the two highlighted
simulations in figure 8. This comparison illustrates the fact that simulations with strong
reductions in the dry matter allocation function show correspondingly small reductions in the



stress function regulating transpiration or, as in this example (simulation number 6), none at
all. This is because an increased rate of leaf loss and a greater allocation of assimilates to the
below-ground biomass during drought reduces the transpiration demand as well as increasing
the potential rate of water uptake by the root system. These adaptation mechanisms in
response to soil drying conserve soil water and reduce the likelihood of stomatal closure, so
that transpiration can be maintained during extended dry summer periods at Selhausen.

**3.3 Grassland growth**


Figures 9 and 10 show comparisons of the acceptable simulations with the measurements of
leaf area index and harvested biomass on the lysimeters at Selhausen and Rollesbroich. The
model efficiencies for these two data types are shown in table 6. Figure 11 shows box and
whisker plots of the simulated total harvest and overall water use efficiencies (WUE, defined
as total harvest divided by evapotranspiration) at the two sites. The results suggest that the
model performed satisfactorily for leaf area development at both sites and for harvested
biomass at Selhausen, but not for harvests at Rollesbroich (table 6). These poorer results can
largely be explained by the fact that lysimeter Ro_Y_013 was considered an outlier, so no
effort was made to match this data by loosening the constraints in the GLUE analysis.
Figure 12 shows the gain and loss terms in the dry matter balances simulated with the 30 best
parameterizations at each site. Simulated assimilation was ca. 10% larger at Selhausen
compared with Rollesbroich as a consequence of the greater radiation input and higher
temperatures (Figure S1) and the fact that water stress is only slightly more prevalent (Figure
8). Leaf loss is a relatively small term in the mass balance (10-12% of assimilation) and is similar
at both sites (Figure 12). Root production and decay (i.e. turnover) are more significant terms,
with root decay closely mirroring production, since it is modelled as a first-order function of
biomass. Expressed as a proportion of assimilation, simulated root production and decay is
somewhat larger at Selhausen compared with Rollesbroich (ca. 58 and 53% of assimilation
respectively, on average, for both), while root biomass is also somewhat larger at Selhausen
(see figure S8). This is in agreement with experimental studies that have demonstrated
increases in below-ground biomass production in grasslands as a consequence of drought (e.g.
Jones et al., 1980a; Kahmen et al., 2005; Wedderburn et al., 2010; Skinner and Comas, 2010;
Padilla et al., 2013; Nosalewicz et al., 2018; Meurer et al., 2019). It was not possible to make
measurements of root biomass and production in the lysimeters at the two sites due to the
constraints of the experimental set-up. However, literature data on root biomass and
production in similar temperate grassland environments can serve as an approximate "reality-
check", suggesting that our simulations (Figure S8) are reasonable. For example, in northern
Germany, Chen et al. (2016) measured a root biomass of ca. 500 g m$^{-2}$ at 0-30 cm depth and a
growth rate of 450 g m$^{-2}$ year$^{-1}$, while in central Sweden, Meurer et al. (2019) found a root
biomass of 250-330 g m$^{-2}$ in the same depth interval. In central France, Picon-Cochard et al.
(2012) reported summer peak root biomasses of 13 perennial grasses grown in monoculture
varying between ca. 400 and 800 g m$^{-2}$, with a temporal pattern matching that simulated by
our model (Figure S8). Likewise, Wedderburn et al. (2010) reported peak root counts in early
summer and a minimum in winter for *Lolium perenne* pastures in New Zealand. The values of
below-ground production simulated by our model are also within the range reported by Hui
and Jackson (2006) for temperate grasslands in a global meta-analysis.
## 4. Conclusions
In this study, we made use of an eco-hydrological model to analyze the impacts on soil water
balance and grassland production of climate change triggered by the transfer of weighing
lysimeters from a wet, cool climate (Rollebroich) to a drier, warmer climate (Selhausen). The
relatively simple model employed in this study gave excellent simulations of soil water
contents (Model Efficiency, ME, between 0.24 and 0.87) and evapotranspiration rates (ME
between 0.32 and 0.60) measured at a daily resolution at both sites during a six-year period,
as well as acceptable simulations of leaf area development (ME between -0.04 and 0.50). In
this model application, we assumed identical static root distributions for the grassland at the
two sites and inferred different (constant) values of the maximum root depth, with deeper
roots in the drier climate at Selhausen. We also concluded from the modelling that more
frequent and intense soil drying at Selhausen led to a shift towards a greater production of
below-ground biomass. A major challenge for the future will be to further develop crop and
eco-hydrological models to enable them to predict these dynamic responses of plant roots to
changing soil and climatic conditions as emergent phenomena. In this respect, it should be
worthwhile to test simple empirical approaches to link root distribution with maximum root
depth and biomass (e.g. Arora and Boer, 2003) as well as developing improved architectural
models of root growth (e.g. Postma et al., 2017; Schnepf et al., 2018; Mboh et al., 2019).
Regardless of modelling approach, it seems clear that plastic responses of plant traits to
climate change of the kind we inferred from our study (e.g. in root depth or leaf conductance)
introduce significant uncertainties into model predictions of water balance and plant growth.
**Data availability**
The raw data can be freely obtained from the TERENO data portal (https://teodoor.icg.kfa-
juelich.de/ddp/index.jsp). Processed data developed during this study can be acquired upon
request from Jannis Groh or Katharina Meurer.
**Author contributions**
The study was conceived by NJ, HV, KM and EL. NJ built the model. TP, JG, WD and CB supplied
data and advised on its use. Initial data analyses and model applications were carried out by
ER as part of his thesis project, supervised by KM, NJ and EL. NJ and KM carried out the final
simulations. NJ prepared the manuscript with contributions from all authors.
**Competing interests**
The authors declare that they have no conflict of interest.
**Acknowledgments**
This work was partly funded by the Swedish Research Council for Sustainable Development
(FORMAS, grant no. 2018-02319). We also acknowledge the support of the TERENO-SoilCan
program funded by the Helmholtz Association (HGF) and the Federal Ministry of Education
and Research (BMBF). We would also like to thank Werner Küpper, Ferdinand Engels, Philipp



Meulendick, Rainer Harms, and Leander Fürst at the Selhausen and Rollesbroich lysimeter
stations for their support.



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



Table 1. Soil properties at Rollesbroich

| Depth (cm) | Particle size distribution (%), fine earth fraction | | | Texture class (U.S.D.A.) | Organic carbon (%) | pH (CaCl$_2$) |
|---|---|---|---|---|---|---|
| | Clay (<2 μm) | Silt (2-50 μm) | Sand (50-2000μm) | | | |
| 0-7 | 19 | 14 | 67 | Sandy loam | 5.3 | 5.2 |
| 7-24 | 9 | 33 | 58 | Sandy loam | 2.5 | 5.3 |
| 24-42 | 37 | 23 | 40 | Clay loam | 1.2 | 5.4 |
| 42-50 | 35 | 33 | 32 | Clay loam | 0.8 | 5.4 |
| 50-71 | 32 | 32 | 36 | Clay loam | 0.3 | 5.4 |
| 71-93 | 32 | 32 | 36 | Clay loam | 0.3 | 5.2 |
| 93-127 | 17 | 24 | 59 | Sandy loam | 0.1 | 4.6 |
| 127+ | 22 | 30 | 48 | Loam | 0.2 | 4.9 |





Table 2. Measured water balance, harvested biomass and water use efficiency for the lysimeters (annual averages for the period 2013-2018; P = precipitation, PET = potential evapotranspiration calculated with the FAO Penman-Monteith method, AET = actual evapotranspiration, ΔS is the change of water storage calculated as P-AET-Percolation and WUE is water use efficiency defined as harvested biomass (Harvest) divided by AET).

| Site | Lysimeter | P | PET | AET | Percolation | ΔS | Harvest [g DM m$^{-2}$ year$^{-1}$] | WUE [kg DM m$^{-3}$ water] |
|------|-----------|-----|-----|-----|-------------|-----|--------|------|
| | | | | [mm/year] | | | | |
| Rollesbroich | Ro1 | 1055 | | 649 | 438 | -31 | 732 | 1.13 |
| | Ro3 | 1079 | 710 | 651 | 466 | -38 | 907 | 1.39 |
| | Ro5 | 1050 | | 623 | 422 | 5 | 678 | 1.09 |
| | Average | 1062 | | 641 | 442 | -21 | 772 | 1.20 |
| Selhausen | Se21 | 696 | | 716 | -42 | 22 | 691 | 0.97 |
| | Se25 | 690 | 827 | 709 | -58 | 38 | 665 | 0.94 |
| | Se26 | 699 | | 714 | -14 | -1 | 661 | 0.93 |
| | Average | 695 | | 713 | -38 | 20 | 672 | 0.94 |



Table 3. Soil hydraulic parameters used in the modelling

| Depths (cm) | Parameter | | | | | |
|---|---|---|---|---|---|---|
| | $\theta_s$ (m³ m⁻³) | | $\alpha$ (cm⁻¹) | $n$ (-) | $K_{10}$ (cm h⁻¹) | $\tau$ (-) |
| | Selhausen | Rollesbroich | | | | |
| 0-24 | 0.45 | 0.55 | 0.025 | 1.34 | 1.89 | 0.5 |
| 24-48 | 0.39 | 0.39 | 0.025 | 1.09 | 0.73 | 0.5 |
| 48-90 | 0.38 | 0.38 | 0.025 | 1.08 | 0.83 | 0.5 |
| 90-140 | 0.38 | 0.38 | 0.025 | 1.17 | 1.46 | 0.5 |





Table 4. Fixed values for plant parameters at both sites

| Parameter | Value | Sources/comments |
|---|---|---|
| ***Above-ground parameters*** | | |
| Maximum radiation use efficiency, $RUE_{max}$ (MJ m$^{-2}$ d$^{-1}$) | 1.6 | [1]Akmal and Janssens (2004) |
| Leaf loss coefficient, $k_{ag}$ (d$^{-1}$) | 0.02 | Istanbulluoglu et al. (2012) |
| Specific leaf area, $S_{leaf}$ (cm$^2$ g$^{-1}$) | 142 | Site data |
| Base temperature, $T_b$ (°C) for stomatal conductance and assimilation | 0 | [2]Wingler (2015), Körner (2008, 2015) |
| Base temperature, $T_b$ (°C) for DM allocation and leaf loss | 5 | [2]Schapendonk et al. (1998), Black et al. (2006), Hennessy et al. (2008) |
| Optimum temperatures, $T_{o(low)}, T_{o(high)}$ (°C) Ceiling temperature $T_c$ (°C) | 12, 25 35 | Howard and Watschke (1991), Wu et al. (2016), Loka et al. (2019) |
| Limiting soil water pressure head for cessation of transpiration, $\psi_w$ (m) | -150 | Standard assumption |
| Fraction of assimilates allocated to roots under optimal conditions, $f_{bg(opt)}$ (-) | 0.5 | Hui and Jackson (2006) |
| ***Below-ground parameters*** | | |
| Root decay constant, $k_{bg}$ (d$^{-1}$) | 0.007 | Van der Krift and Berendse (2002), Chen and Brassard (2013) |
| Root radius, $r_o$ (cm) | 0.02 | Van der Krift and Berendse (2002), Picon-Cochard et al. (2012) |
| Effective root fraction, $\varepsilon$ (-) | 0.05 | Faria et al. (2010) |
| Specific root length, $S_{root}$ (m g$^{-1}$) | 118 | Picon-Cochard et al. (2012) |
| Shape factor for root distribution, $c$ (-) | -1.2 | Schenk and Jackson (2002), Fan et al. (2016) |

[1] assuming PAR = 50% of incoming solar radiation

[2] transpiration/assimilation is less sensitive to low temperatures than growth



Table 5. Uncertain parameters: initial ranges, data sources and post-priori parameter ranges

| Parameter | Ranges sampled | Post-priori parameter values (n=30) | | | | | |
|---|---|---|---|---|---|---|---|
| | | Selhausen | | | Rollesbroich | | |
| | | Median | Inter-quartile range | $10^{th}$, $90^{th}$ percentiles | Median | Inter-quartile range | $10^{th}$, $90^{th}$ percentiles |
| Radiation extinction coefficient, $\beta$ | [1]0.4-0.8 | 0.57 | 0.51-0.65 | 0.48, 0.71 | 0.58 | 0.51-0.65 | 0.48, 0.71 |
| Maximum stomatal conductance, $k_{sto(max)}$ (cm s$^{-1}$) | [2]0.4-1.6 | 1.28 | 1.13-1.47 | 0.97, 1.56 | 0.60 | 0.48-0.83 | 0.46, 0.96 |
| Maximum root depth, $D_r$ (cm) | [3]40-100 | 79 | 75-83 | 70, 86 | 56 | 48-67 | 42, 73 |
| Limiting pressure head at the root surface, $\psi_{o(crit)}$ (-cm) | [4]100-2000 | 271 | 233-347 | 195, 533 | 271 | 224-347 | 157, 419 |

[1] Schapendonk et al. (1998), Akmals and Janssens (2004), White and Snow (2012), Zhang et al. (2014)

[2] Nijs et al. (1997), Allen et al. (1998), Wang and Huang, (2003), DaCosta et al. (2004), Dong et al. (2011), Holloway-Phillips and Brodribb (2011), Hu et al., 2013

[3] Site observations; Jackson et al. (1996), Schenk and Jackson (2002), Fan et al. (2016)

[4] No information is available, hence a wide 'a priori' uncertainty range was selected



Table 6. Model efficiencies for the different data types (median values of the 30 acceptable parameter sets, with minimum and maximum values in parentheses).

| Site | Model efficiency | | | | | |
|------|------------------|---|---|---|---|---|
| | Water content at 10cm depth | Water content at 30 cm depth | Water content at 50 cm depth | Evapo-transpiration | Harvest | Leaf area index |
| Ro | 0.84 (0.78, 0.87) | 0.77 (0.58, 0.83) | 0.73 (0.64, 0.86) | 0.58 (0.54, 0.60) | -0.70 (-0.54, -0.81) | 0.19 (0.09, 0.50) |
| Se | 0.81 (0.75, 0.84) | 0.68 (0.58, 0.73) | 0.28 (0.24, 0.31) | 0.38 (0.32, 0.45) | 0.35 (0.15, 0.46) | 0.15 (-0.04, 0.32) |





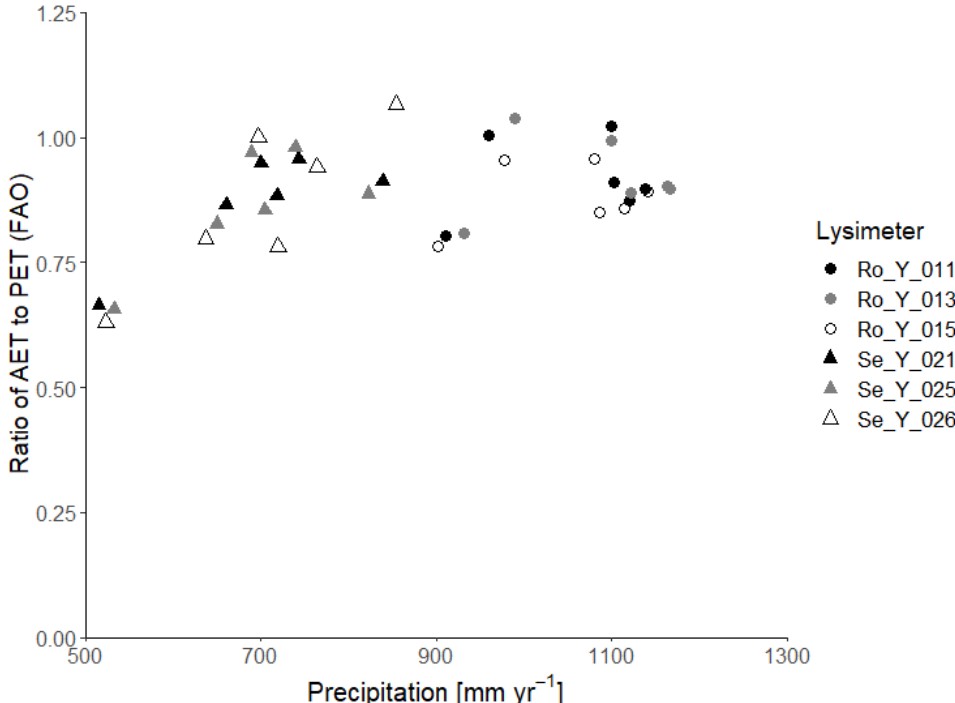

Figure 1. Ratio of actual evapotranspiration (AET) to potential evapotranspiration (PET-FAO) calculated with the FAO Penman-Monteith method (Allen et al., 1998) as a function of precipitation at Selhausen and Rollesbroich on an annual basis for the period 2013-2018.

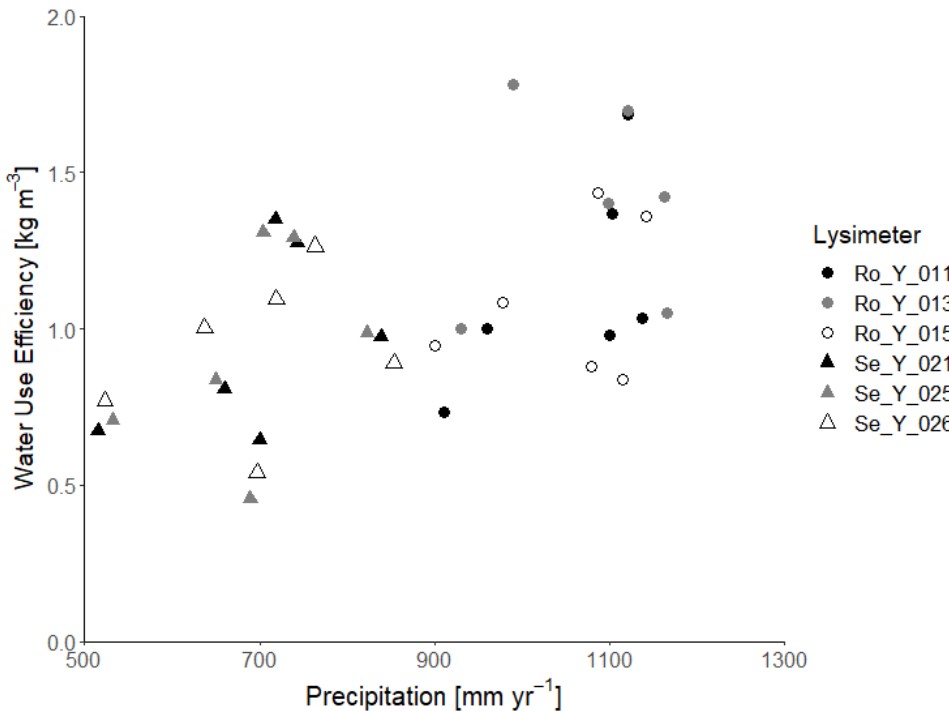

Figure 2. Water use efficiency (= annual harvest divided by annual evapotranspiration) as a function of annual precipitation at Selhausen and Rollesbroich.

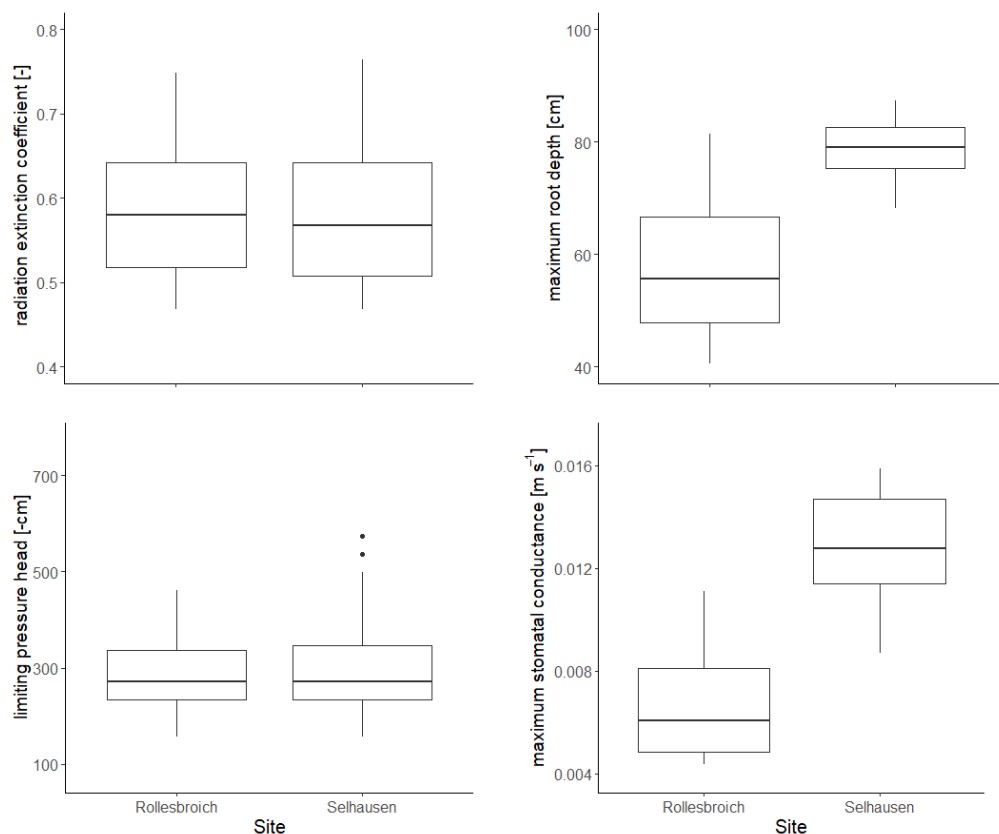

Figure 3. Posterior distributions of the four parameters treated as uncertain in the GLUE analysis. The horizontal line is the median value for the acceptable parameter sets, the box denotes 25th and 75th percentiles (inter-quartile range), the whiskers cover data points that lie within 1.5 times the inter-quartile range and solid circles represent outliers outside this range.



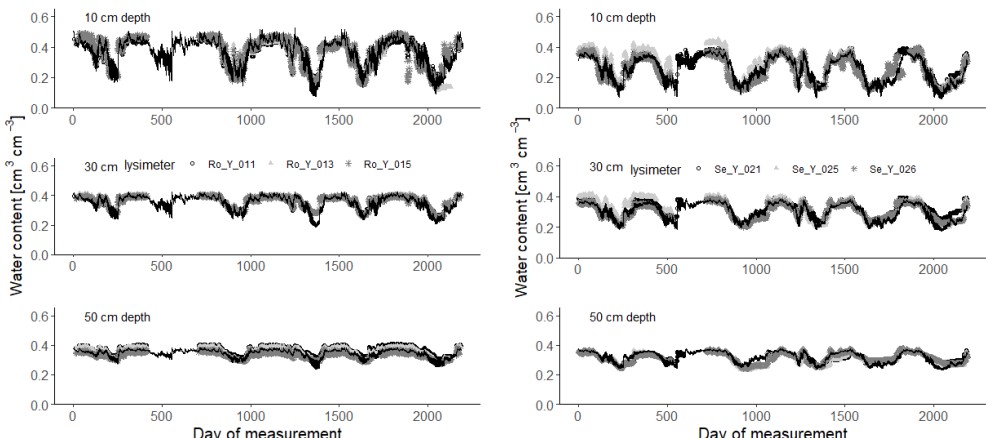

Figure 4. Measured soil water contents (symbols) at 10, 30 and 50 cm depth (2013-2018) compared with simulations for the 30 acceptable parameterizations at each site (black lines).



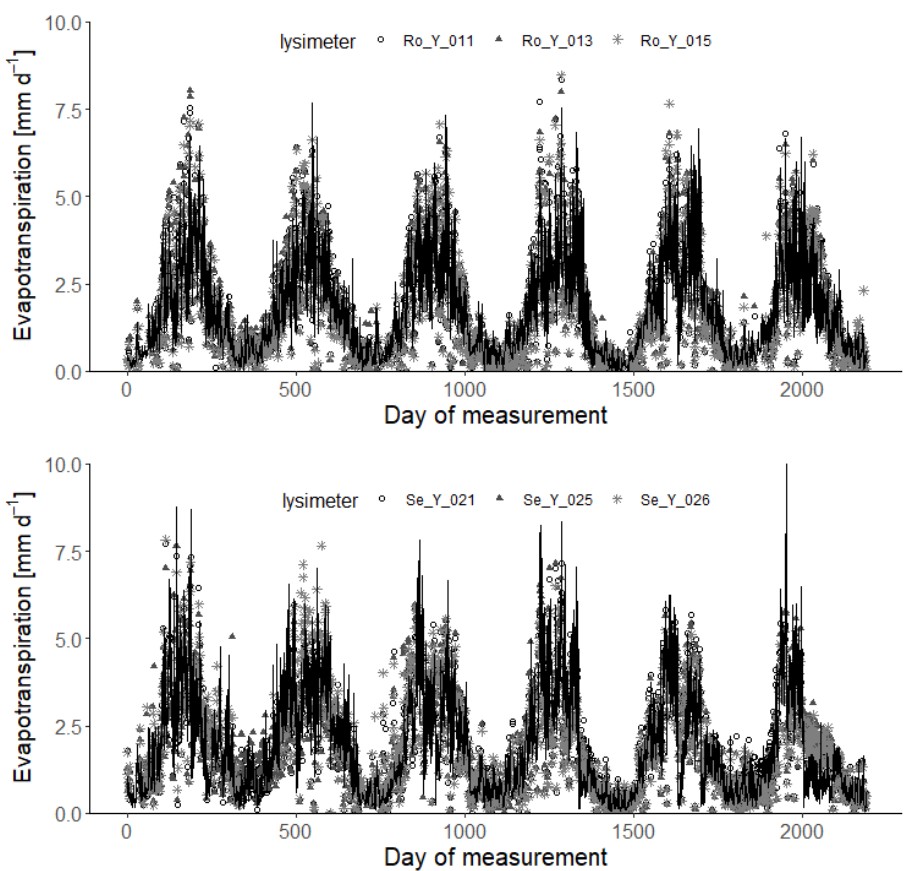

Figure 5. Measured daily evapotranspiration rates (symbols; 2013-2018) compared with simulations for the 30 acceptable parameterizations at each site (black lines).

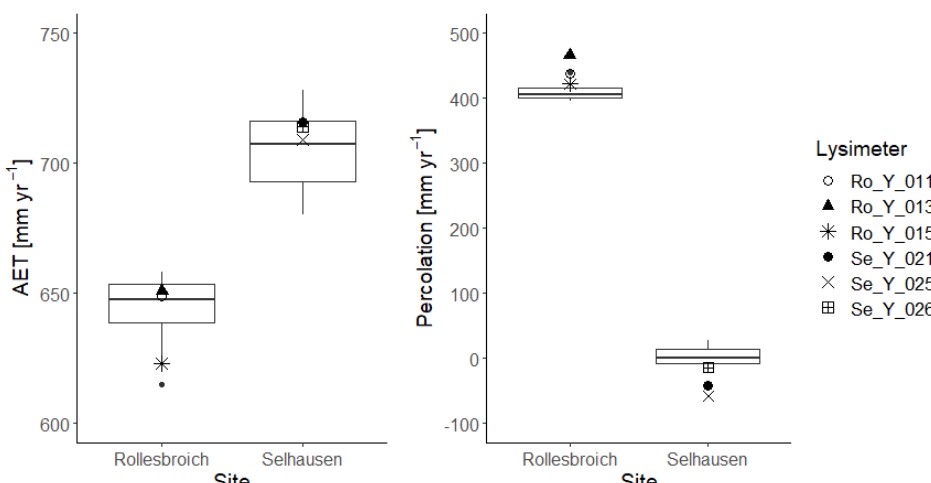

Figure 6. Box and whisker plots of simulated annual average evapotranspiration (AET) and percolation at Selhausen and Rollesbroich for the period 2013-2018 for the 30 acceptable simulations compared with the lysimeter measurements (large symbols). For an explanation of the box and whisker plots, see the caption to figure 3.





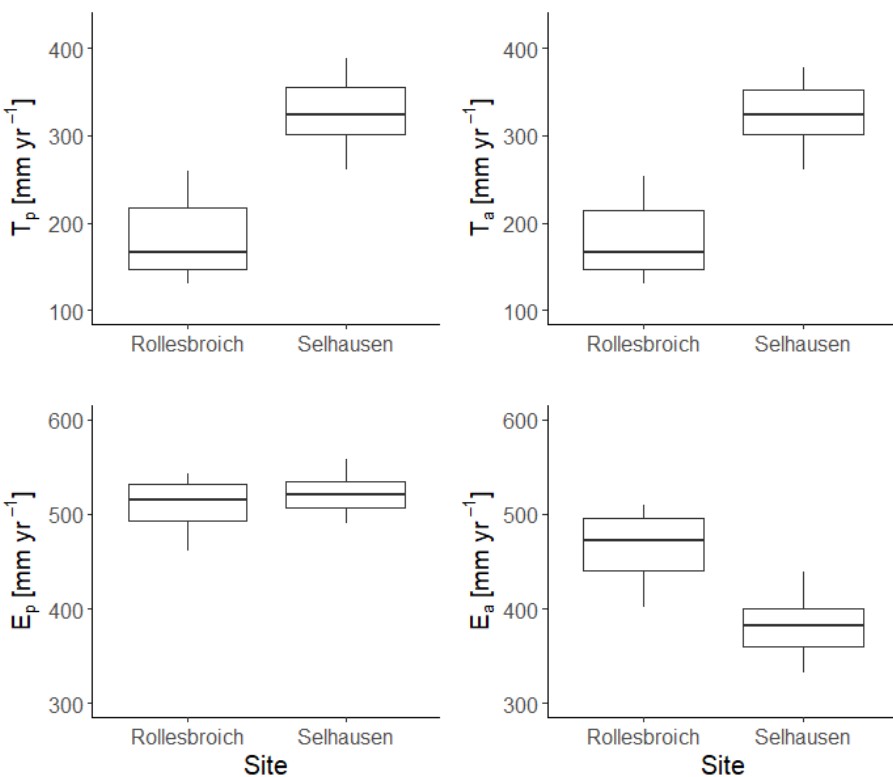

Figure 7. Simulated water balance terms for the 30 acceptable simulations at each site. For an explanation of the box and whisker plots, see the caption to figure 3.

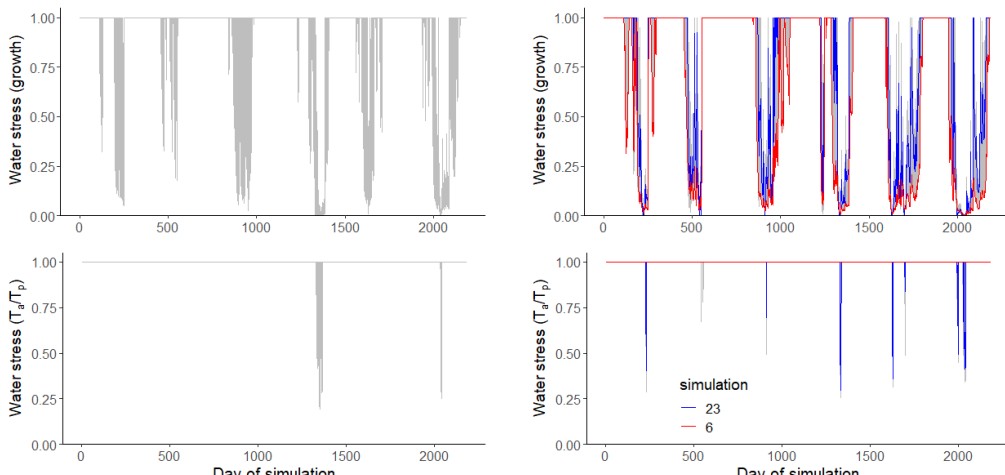

Figure 8. Plots of the two water stress functions in the model for the acceptable simulations. The uppermost figures show the threshold function of the pressure head at the root surface (equation 35) controlling dry matter allocation and leaf loss, while the figures at the bottom show the ratio of actual to potential transpiration, which controls assimilation (equation 34). Two contrasting acceptable simulations for the Selhausen site are highlighted in red and blue.



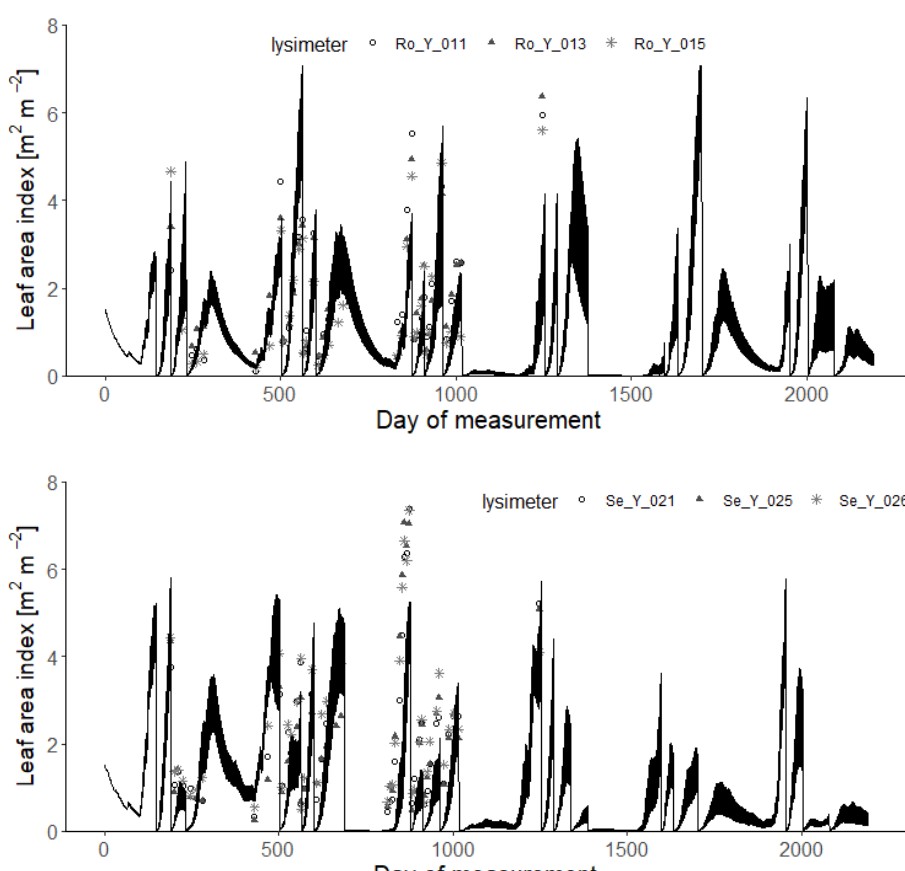

Figure 9. Measured daily leaf area index (symbols; 2013-2018) compared with simulations for the 30 acceptable parameterizations at each site (black lines).

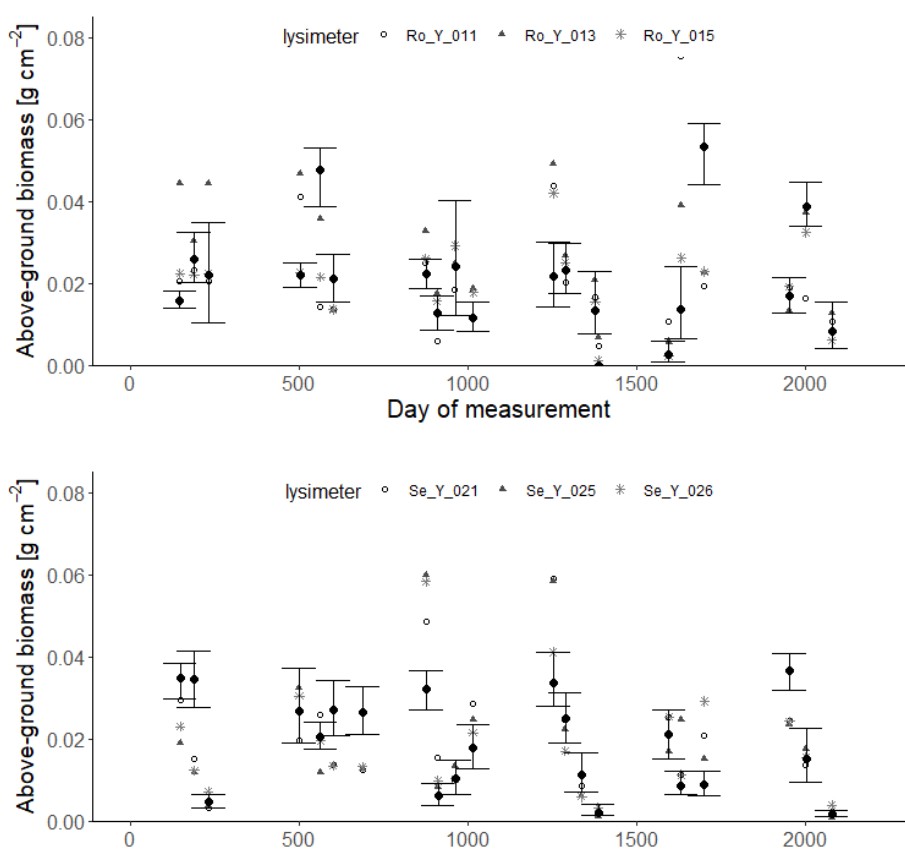

Figure 10. Measured harvests of above-ground biomass (symbols; 2013-2018) compared with simulations at each site (black symbols indicate means of the 30 acceptable parameterizations and the vertical lines denote minimum and maximum values).

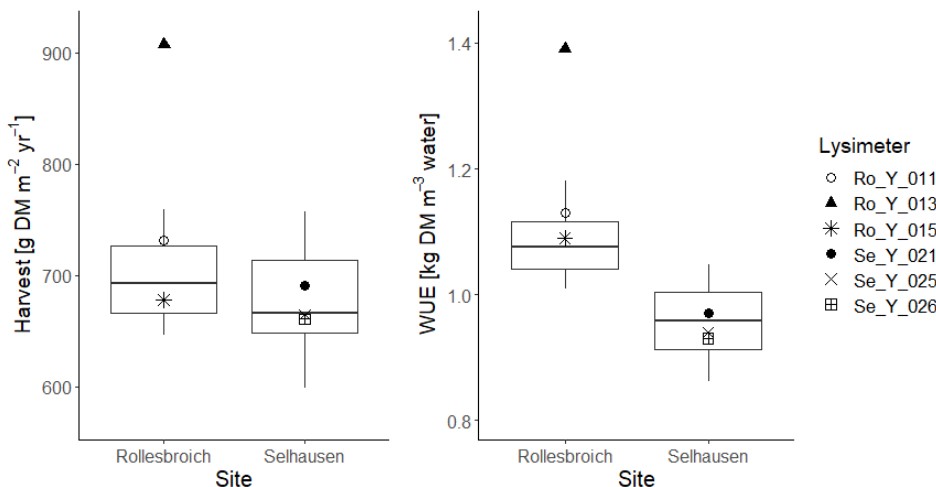

Figure 11. Box and whisker plots of simulated harvests and water use efficiencies (WUE, defined as total harvest divided by evapotranspiration) at Selhausen and Rollesbroich for the period 2013-2018 for the 30 acceptable simulations compared with lysimeter measurements (symbols). For an explanation of the box and whisker plots, see the caption to figure 3.



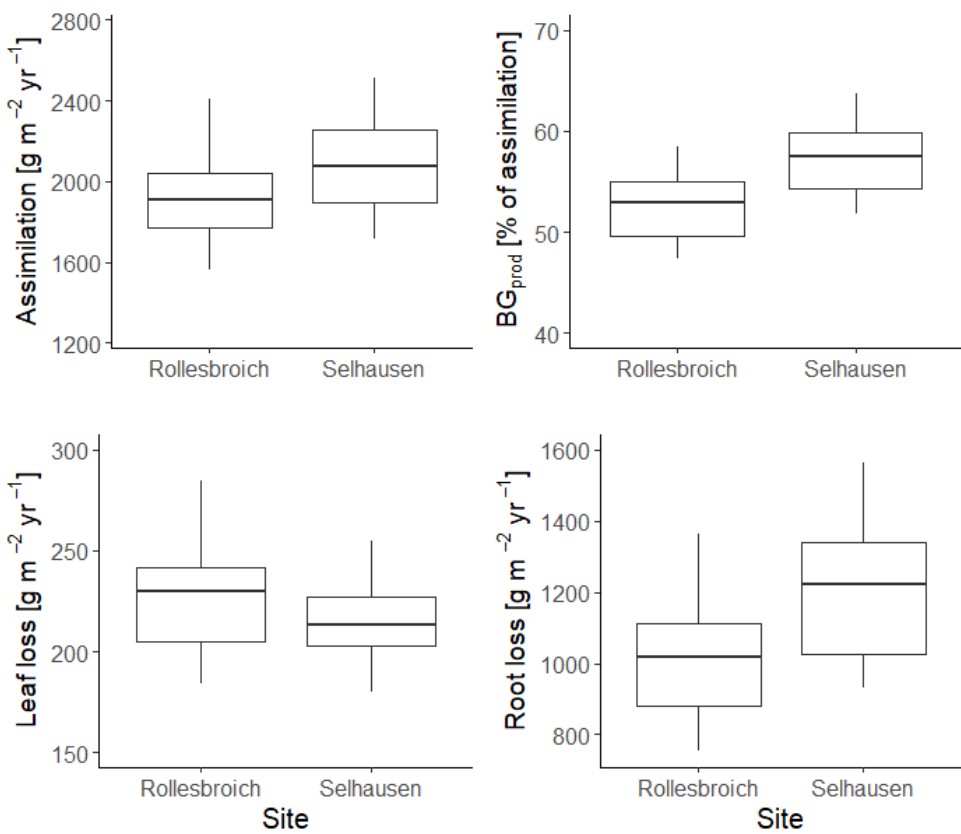

Figure 12. Box and whisker plots showing the simulated terms in the dry matter balance for the 30 acceptable model parameterizations at Selhausen and Rollesbroich for the period 2013-2018. For an explanation of the box and whisker plots, see the caption to figure 3.