# Peer review of "Coupled modelling of hydrological processes and grassland production in two contrasting climates"

_Hydrology and Earth System Sciences, 2021_

## Author Comment (AC1)

Referee comment on "Coupled modelling of hydrological processes and grassland production in two contrasting climates" by Nicholas Jarvis et al., Hydrol. Earth Syst. Sci. Discuss., https://doi.org/10.5194/hess-2021-316-RC1, 2021

This manuscript describes an interesting modelling study of soil water balance components in lysimeters with grassland production, including the simulation of cutting/grazing, performed in two contrasting climate zones in Germany. Six lysimeters were collected in a wetter region (Rollesbroich), three of them were transported to a drier zone (Selhausen) to allow studying the same soil under different climate conditions. This practice, here called "space-for-time substitution", could be helpful to mimic future climate conditions.

For the modelling, a Richards equation-based hydrological model is developed, using Van Genuchten-Mualem soil hydraulic parameters. Special attention is given to root growth and distribution, and a process-based function is used to predict root water uptake as a function of depth, and actual transpiration rates. Potential grassland growth (dry matter accumulation) is modelled using the concept of RUE. Actual growth rates are calculated from potential rates by reducing for water and temperature stresses. The effect of water stress on dry matter accumulation is included by assuming a linear proportionality between relative transpiration (predicted by process-based root water uptake modelling) and relative growth. Temperature stress is added by establishing a piecewise linear function with zones delimited by specific threshold temperatures. Simplifications, especially with respect to root growth and distribution, are unavoidable in this kind of modelling approach and are well presented and justified. The manuscript is generally well written. There are some issues to be addressed by the authors, among others referring to the soil hydraulic parameterization and clarity about the lysimeter soil contents. See my specific comments below.

We would like to thank the referee for the many constructive comments on our manuscript. We agree with nearly all of these suggestions (see below) and we are sure that this will significantly improve the paper.

Besides these, in my opinion, *the most important shortcoming (making me suggest a major revision) of the manuscript refers to the calibration and validation procedure*. You used the GLUE method to identify the best parameterizations using 6 years of lysimeter observations. You then discuss the model performance based on the 30 best parameter combinations selected by some criteria out of 2000 original combinations (parameter realizations). Posterior results seem fairly good, in terms of soil water content and ET (Figs 4 and 5), as well as model efficiencies (Table 6). But isn´t that to be expected when selecting the 30 best performing parameter sets? I would challenge you to follow a more rigorous calibration-validation protocol, performing the GLUE method on three or four years of your data, selecting the best parameter combinations, and then testing them on the remaining two or three years. This would reveal an unbiased and much more convincing model performance.

We realize now that our objective with the modelling was not very well explained at the end of the introduction at lines 118-126, as pointed out by the referee in a later comment. We will modify this part of the text to make the objective of the modelling exercise much clearer.

The main aim of the modelling exercise in our paper was actually not really to try to "validate" a model, but rather to use the model to investigate likely or plausible reasons for the differences in hydrological and plant response at the two sites. As the referee suggests, the (mostly) acceptable model efficiencies give us confidence that our model is good enough for this purpose. To answer the question posed by the referee: no, we doubt that we would get good results from taking the best 30 simulations of 2000 parameterizations of any given model. We would not expect good results if the model was inadequate. To give an example: we have also run calibrations for a model which is identical in all respects to the one described in our paper, except that it does not account for compensatory root water uptake (the Feddes root water uptake model is used instead). This model performs badly, with large negative model efficiencies for soil water contents and water balance terms, even for the best 30 simulations. This contrasting outcome emphasizes that model "validation" is really only a worthwhile exercise in a relative sense i.e. comparing the performance of different models.

From a practical point of view, the length of our time series (6 years) is anyway much too short in relation to the variations in weather from year to year to make it worthwhile to carry out a split validation. If we applied GLUE on the first two or three years of data only, we would not be including years with very low precipitation. This would mean that parameters related to drought response of the grassland would not be very well identified, due to their lack of sensitivity. In other words, we would get wider bounds for the acceptable parameters, if we applied the model only to the first 2 or 3 years. Many of these acceptable parameter sets would then be considered unacceptable in the remaining years, since they would fail to predict correct responses in the very dry years that were not represented in the first period. A very small proportion of parameter sets would probably be acceptable for both periods. But to get a sufficient number of these acceptable parameterizations for both periods would require very many more simulations (which would just not be practicable). In summary, we can use the model to learn from the data in the most efficient way by applying the GLUE procedure for the entire 6-year period. In fact, split validation would almost certainly not be a good approach even if our time series had been much longer than 6 years. The inefficient and potentially highly misleading nature of the split-sample validation procedure has been demonstrated, for example, by Arsenault et al. (2018, Journal of Hydrology, 566, 346-362) who used a time series of 16 years and came to the conclusion that split validation is …

*… futile at best, detrimental at worst, and deceiving in all cases …*

The conclusion section contains some discussion but does not mention the interesting fact that plants were able to mitigate water stress in the drier climate by enhancing below ground (root) growth. If this is confirmed, soil organic matter contents could be expected to rise when climate change triggers more water stress. Also, in agricultural (grain) crops, above ground yields might then be expected to diminish to favour belowground development. This is important in several ways, one of them being the expected yield of agricultural crops.

Yes, we certainly agree that this is an interesting and important result, but we did in fact already mention this in the conclusions (at lines 699-700). Related to the last part of this comment, it is worth mentioning that another study with the SoilCan lysimeters containing arable crops has shown the opposite effect i.e. that in the drier climate, water use efficiency increased, with larger grain yield per unit of water consumed. We suppose that putting more

resources into seed production is probably a useful evolutionary adaptation for annual crops, whereas it seems that perennials try to survive droughts by enhancing root growth.

Scientific questions/issues:

l118-126 This final part of the introduction appears to refer to Materials and methods. I suggest finalizing the introduction with a concluding statement (possibly containing the objectives), and start the M&M section with the content of these lines. Related to this, please state more clearly that the six lysimeters contain soil from Rollesbroich, but that three of them were transferred to Selhausen. l162 somehow contains this information, but it should be stated more clearly. The sentence in l179-181 is clear with this respect but should be located at an earlier position in the text.

Yes, we agree, and we will modify the text accordingly

l194-196 This net upward flow would imply a depletion of the groundwater body in the long term? I.e., if plants invest more in the belowground biomass to avoid drought stress, large-scale hydrological cycles would be affected? A comment in this respect seems appropriate here.

Yes, actually, the Selhausen site is located on a low-lying relatively flat flood plain of the river Rhine (lines 130-131) and so it seems probable that this net upward flow is sustained by groundwater flow from the surrounding hills. In other words, the site lies in a discharge area (rather than a recharge area). We agree that we should add a sentence or two to explain this.

l210-213 Here you suggest that yield may have been affected by nutrient deficiencies. The performed simulations assume optimum plant nutrition (l356). Shortly discuss if this would affect the fairness of assessing model performance based on experimental data including nutrient issues.

Yes, it would be more accurate to write that the model does not account for variations in plant nutritional status. We will modify the text at line 356 accordingly.

l284-294 No details are given about how you (numerically) solved the Richards equation (13). Was an implicit algorithm scheme used? How about the discretization? Time and space steps? Averaging K and h? Some more details would be important.

Yes, we agree. We will add these details.

l429 Regarding "the fraction of the total root length that is effective for water uptake" (epsilon), you use a fixed value of 0.05 (after Faria et al., 2010). A high model sensitivity to this parameter is to be expected, and it is also plausible to assume that epsilon will increase in periods of more intense root growth and decrease when the root system is shrinking due to stress. Please discuss shortly the effect of assuming a fixed static value for this parameter.

Yes, our sensitivity analysis showed that the effective root fraction is indeed a moderately sensitive parameter (see table S2). However, for practical reasons, we could not include a large number of parameters in the uncertainty analysis. We selected four important parameters to calibrate and the reasons for this choice are already discussed at lines 518-533. One reason for not selecting the effective root fraction is that it would likely be correlated with other sensitive plant parameters, in particular with $f_{bg\ (opt)}$ and $\psi_{o(crit)}$. This is described at line 533.

l481-495 There are a lot of assumptions, guesses, and uncertainties in the soil hydraulic parameterization described here.
Perhaps, but we believe that we have more data on which to base our parameterization than is usually the case in modelling studies of this kind.

It is especially peculiar that the saturated water content of the surface layer is much higher in the lysimeters at Rollesbroich, as "estimated from the data by eye". Assuming all lysimeters contain a similar soil monolith (collected at Selhausen), can this be made plausible?
Yes, this is interesting. This could be the result of chance spatial variation. But it is also possible that the physical properties of the uppermost layers of the Rollesbroich soil have changed following the move of the lysimeters to Selhausen. One plausible explanation is that the drier soil conditions at Selhausen have led to increased mineralization rates of soil organic matter, leading to a decline in soil organic matter content and consequently, an increase in bulk density (i.e. a loss of porosity). It is also possible that the drier soil conditions at Selhausen have induced water repellency, which reduces soil wettability.

We will add some text on this in the revised version.

In l468-469, the reader is informed that a common parameterization for the soil hydraulic properties in all six lysimeters will be assumed. But apparently, this does not include ThetaS?
Yes, this is true. It is only common among the three replicates at each site. We will change this statement accordingly.

Please improve the description/justification of the soil hydraulic parameterization.
Yes, we will add some further explanation and modify existing statements appropriately

l582 "No differences between the two sites were found for two of the parameters" – this is an interesting result and merits more attention. I think it is corroborated by common sense. Both parameters (the radiation extinction coefficient and the parameter controlling dry matter allocation and leaf loss as a function of water stress) are expected to be mostly genotypical and not affected by environmental conditions. On the other hand, unstressed stomatal conductance (discussed in l591-602) is expected to be a function of sink (atmosphere) conditions, especially VPD, temperature, radiation; rooting depth will probably be affected by soil moisture and temperature distribution in the soil profile, both a function of weather conditions.
These are very interesting observations. We will add some text along these lines.

l627 Here one of the reasons for the high E values at Rollesbroich is assumed to be "the capillary nature of the soil". This is a too vague description. Soils are very similar at both locations. I suggest you remove this reason.

Yes, we agree. We will delete this.

l700-705 ("A major … growth") These statements are no conclusions from the presented research, they would fit better in the Discussion than in the Conclusion section.
We don't agree. These are conclusions that we can draw from the results of our study.

Some other minor "technical corrections":

l62 SVAT stands for "Soil-Vegetation-Atmosphere Transfer". Please add "Transfer".'
Yes, OK, we will
l79 Johnson et al. (2008) would fit well in this list of citations (doi:10.1071/EA07133)
Yes, we can cite this paper
l121 Delete "In this study".
Yes, O.K.
l235-240 For clarity, please add the units (dimensions) of the parameters of eqs. 1-7 in these lines.
OK, yes, we will do so
l270 To be more precise, one would need pressures and conductances, both unavailable in bucket models.
Yes, true, we will add this
l288 (here and on other occasions) The unit "day" is officially abbreviated as d. I suggest using d instead of day, days throughout the text.
OK, we will do so
l306 I would prefer to say that tau (just like alpha and n) is a (fitting) shape parameter. The way you wrote it here, it seems tau could be independently measured.
We don't think that what we wrote here implies anything about the way these parameters can be estimated. But we will try to write this in an even more neutral way.
l700 after "below-ground biomass" you might add: "thus mitigating drought stress".
Yes, we can do this
Figs. 1 and 2, X-axis label and Figs. 6, 7, 11, and 12, Y-axis: replace yr-1 by y-1
We would prefer not to (but we will do so if the editor insists)
Fig. 4 Symbols in this figure can hardly be distinguished, I suggest using colours for the three lysimeters. It would also be good to clearly identify both columns of figures by Selhausen and Rollesbroich (they are now identified only by the codes of the respective lysimeters).
Yes, we will add the site names to the columns of figures. But adding colours to the symbols does not help. There is so much data (daily time resolution) and the replicates are so close to one another, that the symbols will be indistinguishable regardless of what we do.
Figs. 4, 5, 8, 9, and 10: X-axis label is unclear. I would prefer "Time from onset of simulation (01-Jan-2013) [d]"
Yes, we agree that it is not 100% clear at present. But rather than change the x-axis label, we prefer to add some clarifying information in the caption i.e. Day 1 = 1$^{st}$ January 2013
Figure 7: Interpretation would be easier if Tp and Ta (and Ep and Ea) would appear in the same figure, side by side per location. In the current version of the figure, it is difficult to detect any difference between Tp and Ta.
Yes, we can do this. But actually, the differences between Ta and Tp are very small, as we discuss in the text in the following paragraph in relation to figure 8.

---

## Author Comment (AC2)

**Major comments**

We would like to thank the referee for the many constructive comments on our manuscript, which will help us to significantly improve the paper.

This is an interesting and very well written paper, that introduces an novel approach, that of transferring intact lysimeters from a wet cool climate to a much drier and somewhat warmer climate in Germany, to study the effects of climate (change) on the water balance and growth of grass vegetation. The paper has long introduction and materials and methods sections that already reveal a substantial amount of results and bring in a lot of literature references. At times it feels more like a review, and some of these references could perhaps have been saved for the discussion. The methods used are overall sound, although it was not fully clear to me why they had not selected a more recent and complete grass hydrology/growth model.

We had a rather specific purpose with the modelling. We built a relatively simple model to try to understand the reasons for differences in the water balance and grassland growth found at the two sites. Existing crop models include many other processes that are not relevant to our question, and using them would also have been much less transparent.

We will make this objective clearer in the revised version.

Why was interception not included? This would not have caused much computational burden.

It didn't seem to be necessary to account for interception. As noted in the paper, the net evaporative loss caused by interception is usually quite small for short vegetation, since transpiration is reduced by a nearly equivalent amount.

What frustrated me a little was the fact the Results and Discussion section is rather short. After all this pre-amble on how the model works, the reader is still left with quite a few questions.

For example, what are the reasons for the upward flow in the drier lysimeters, even though LAI and Dry Matter are lower? Is this real? Is it the deeper root depth for these lysimeters that could have caused it. I would like to see the authors elaborate a bit more hear, for example by going back to the findings of their sensitivity analyses.

Yes, the net upward flow at Selhausen is definitely real (it has been measured). Yes, the deeper roots may contribute to this, together with the hydrogeological setting and the topographical position of the site on a low-lying flood plain. It seems likely that lateral groundwater flow from the surrounding higher land maintains the supply of water to the drying root zone.

We will add some text to elaborate on this.

**Minor comments**

Line 195-196: "It is also striking that the actual evapotranspiration slightly exceeds precipitation at Selhausen, so that the net percolation at the base of the lysimeters is negative (i.e. an upwards directed flow; Table 2)."…. Does this mean that the percolation was not measured separately? You say that the lysimeters enable the measurement of a complete (closed) water (line 113)? What is causing this negative percolation (capillary rise?).

Percolation was measured separately. We will make this clearer in the revised version (at line 173). The reasons for the negative percolation are mentioned in our response above under "Major comments". We will elaborate on this in the revised version.

Chapter 2, Materials and Methods already contains a lot of results on the lysimeter water balance, AG biomass etc. Should that be moved to the results?

We prefer to keep this in the M&M section, as it sets the scene for the model development (why we model)

Line 214: water use efficiency (WUE) of the grassland in the drier climate was lower than that of the wet climate. Would we have expected the opposite (plants becoming more efficient under dry conditions)? So, is it really the leaf-level WUE hat has changed, or is this the result of other factors?

Yes, a lot depends on how WUE is defined. Note that we defined it from the data we have as the harvested yield divided by total evapotranspiration. The modelling that is described later explains the decrease in WUE as a consequence of an increased allocation of assimilates to the root system and a reduction in above-ground growth. So, this is not leaf-level WUE.

It also seems to depend on the crop in question. We identified for arable lysimeters with a slightly different definition of WUE, that WUE increases after a move to a drier climate (see Groh et al. 2021: https://hess.copernicus.org/articles/24/1211/2020/

How exactly is infiltration calculated in the model?

This is explained at lines 292-297

And the flow at the bottom boundary of the soil profile?

This is explained at lines 290-292

What about Runoff?

There was no surface runoff, as the soil infiltration capacity was never exceeded. We will add a sentence at line 297 to explain this.

In Line 424 and further you talk about feedbacks from the plant growth model to the hydrological model. You mention the effect of LAI and height on the aerodynamic

resistances and hence on the ET fluxes. Surely LAI also affects radiation extinction, and therefore the energy available for ET. This could be mentioned too?

Yes, this is correct. We will mention this in the revised version

Also, the aerodynamic effects will have been relatively low. From that point of view would it not have been better to also consider interception (as it is also affected by LAI)? Although the values for grass are low, they are comparable to winter values of ET, and the equations required are straightforward?

It didn't seem to be necessary to account for interception. As the referee notes here, the net evaporative loss in interception is usually quite small for short vegetation.

Lines 489-492 You say: "The measurements from the matric potential sensors installed in the uppermost soil horizon (0-24 cm depth) appeared to be unreliable. We therefore also used the HYPRES pedotransfer functions to estimate the shape parameter n in the topsoil, while $\alpha$ was set equal to the same value as the deeper horizons. First of all: why where these measurements unreliable? Was the soil too dry?

The pressure potential at 10 cm depth was measured by an MPS 1 sensor and not as in the other depths, by tensiometer. The MPS 1 sensor is relatively good at measuring pressure heads in dry soil, but in comparison with classical tensiometers, it is known to give unreliable values for the range between -200 cm until saturation. The problem here is that this range of pressure heads is important for the definition of the Mualem-van Genuchten parameters (especially theta s and alpha). Thus, in this study we didn't used data from the MPS 1 sensor at 10 cm depth, and relied on the available tensiometer data.

How do you know that the deeper sensors could be deemed reliable?

This type of sensor was not installed in the deeper layers (only in the topsoil)

Also, can I ask why you did not use the VG parameters for the medium-layers where you did have measurements, instead of having to revert to generic HYPRES PTFs? Were these horizons too different?

Yes, that's right. There are large differences in texture between horizons (see lines 141-143)

Your alpha parameter in Table 3 appears to be the same throughout the entire profile, yet n varies considerably. Is this realistic?

Yes, it appears to be reasonably realistic (see figure S3 and table S1)

How come that theta_s in the first soil layer is so much higher in Rollesbroich?

Yes, this is interesting. This could be the result of chance spatial variation. But it is also possible that the physical properties of the uppermost layers of the Rollesbroich soil have changed following the move of the lysimeters to Selhausen. One plausible explanation is that the drier soil conditions at Selhausen have led to increased mineralization rates of soil organic matter,

leading to a decline in soil organic matter content and consequently, an increase in bulk density (i.e. a loss of porosity). It may also be the case that the drier surface soil conditions at Selhausen have reduced soil wettability.

We will add some text in the revised version on this.

In Table 5 you talk about post-priori parameter ranges, whereas in the text (line 579-580) you mention that the posterior uncertainty ranges are much smaller than the prior uncertainty ranges. In figure 3 you talk about posterior distributions of the four parameters. Where exactly are you showing the prior uncertainty ranges? I find this all a little confusing.

We apologize for the confusion. We will change post-priori in table 5 to posterior.

The prior uncertainty ranges are shown in the first column of table 5.

Lines 595-596: You say: "The simulations suggest that the maximum root depth at Selhausen has increased to ca. 80 cm,while the maximum stomatal conductance has roughly doubled". Were you not able to measure the root depth? I guess this would have caused destruction of the lysimeter core.

That's right. We would need to destroy the lysimeters to measure the root depth. This would not be in agreement with the goals of TERENO-SOILCan to provide long-term observations. However, we did observe the initial root depth at the time of sampling (see lines 147-149)

Also, you say that stomatal conductance has doubled, but could it be that the parameter had assimilated aerodynamic effects to changes in vegetation structure? You hint at this perhaps in the following sentences, but it is not clear.

This seems unlikely. Aerodynamic resistance is calculated from plant height, which in turn is estimated from LAI. No differences in the relationships between plant height (and above-ground biomass) and LAI are apparent at the two sites (see figure S6), which suggests that the vegetation structure is similar.

In Figure 4 you need to make it clear which set of 3 graphs is representing which site (the same goes for Fig. 5). It is hidden in the legend but should be more explicit.

Yes, we will do this

Also, based on these plots it is surprising that you opted for a theta_s of 0.55 for Selhausen (estimated "by eye"). I understand that these are daily averages (?), but during the winter months there much have been saturated conditions?

This is a misunderstanding. Theta_s was 0.45 at Selhausen not 0.55 (see table 3)

Water contents do get close to saturation during winter, yes (see figure 4)

In Figure 5 it is quite hard to see what is going on. Would it be possible to separate the years somehow? Or perhaps make cumulative plots?

Yes, we can include cumulative plots (in supplementary)

Line 611-612: You say that the "model performs very well, matching the temporal dynamics in the high-time resolution data on state variables and fluxes as well as reproducing the differences in the overall water balances at the two sites". I am not sure that Table 6 reflects that statement? While model efficiencies are high the values for ET are much lower and values for LAI are negative (with LAI being a crucial variable in many equations), which makes me wonder whether Figure 5 hides a multitude of sins..?

This statement only refers to the hydrological simulations (not the simulation of LAI). Figures 4 to 6 and table 6 undeniably show that the statement is justified.

I find the discussion around Fig. 7 somewhat incomplete. While soil evaporation clearly depends on soil moisture content, windspeed etc. it also depends on incoming radiation (which would have been lower in Ro and higher in Se), and radiation reaching the soil through the canopy. Seeing the DM was lower in Se (and therefore LAI, see also figure 9) I would not necessarily have expected soil evaporation to have been lower for the Se site.

Yes, it is interesting that evaporation was apparently smaller at Selhausen despite higher radiation inputs. We will add some text on this point

Line 628: You say: At Rollesbroich" grassland is harvested 3-4 times during the growing season" Was this not the case at the Selhausen site? This makes comparison between the two sites difficult?

The grassland is also cut at Selhausen 3-4 times per season (see lines 150-153)

The discussion around water stress, with a focus on Figure 8, seems to ignore the fact that one of your earlier figures indicated that the rooting depth at Selhausen was much deeper (80 cm) than at Rollesbroich. Is that mot the main reason for the relatively modest water stress experienced at this site?

Yes, this is good point. We will add this to the discussion

Line 692-693: Are some of these ME values less than excellent if the best value is 1? Can you provide a scale for what consitiues poor, good, excellent etc. in the methods where you introduce ME?

Yes, we agree that this description is not really warranted. We will change "excellent" to "satisfactory". With ME, there is only one objective cut-off: an ME value of less than zero defines poor simulations, see line 559.

---

## Author Response (AR1)

**Responses to referee 1**

We would like to thank the referee for the many constructive comments on our manuscript. We agree with nearly all of these suggestions (see below) and we are sure that this has significantly improved the paper.

This manuscript describes an interesting modelling study of soil water balance components in lysimeters with grassland production, including the simulation of cutting/grazing, performed in two contrasting climate zones in Germany. Six lysimeters were collected in a wetter region (Rollesbroich), three of them were transported to a drier zone (Selhausen) to allow studying the same soil under different climate conditions. This practice, here called "space-for-time substitution", could be helpful to mimic future climate conditions.

For the modelling, a Richards equation-based hydrological model is developed, using Van Genuchten-Mualem soil hydraulic parameters. Special attention is given to root growth and distribution, and a process-based function is used to predict root water uptake as a function of depth, and actual transpiration rates. Potential grassland growth (dry matter accumulation) is modelled using the concept of RUE. Actual growth rates are calculated from potential rates by reducing for water and temperature stresses. The effect of water stress on dry matter accumulation is included by assuming a linear proportionality between relative transpiration (predicted by process-based root water uptake modelling) and relative growth. Temperature stress is added by establishing a piecewise linear function with zones delimited by specific threshold temperatures. Simplifications, especially with respect to root growth and distribution, are unavoidable in this kind of modelling approach and are well presented and justified. The manuscript is generally well written. There are some issues to be addressed by the authors, among others referring to the soil hydraulic parameterization and clarity about the lysimeter soil contents. See my specific comments below.

Besides these, in my opinion, *the most important shortcoming (making me suggest a major revision) of the manuscript refers to the calibration and validation procedure*. You used the GLUE method to identify the best parameterizations using 6 years of lysimeter observations. You then discuss the model performance based on the 30 best parameter combinations selected by some criteria out of 2000 original combinations (parameter realizations). Posterior results seem fairly good, in terms of soil water content and ET (Figs 4 and 5), as well as model efficiencies (Table 6). But isn´t that to be expected when selecting the 30 best performing parameter sets? I would challenge you to follow a more rigorous calibration-validation protocol, performing the GLUE method on three or four years of your data, selecting the best parameter combinations, and then testing them on the remaining two or three years. This would reveal an unbiased and much more convincing model performance.

Re. the question of split validation: the length of our time series (6 years) is much too short in relation to the variations in weather from year to year to make it worthwhile to carry out such an exercise. If we applied GLUE on the first two or three years of data only, we would not be including years with very low precipitation. This would mean that parameters related to drought response of the grassland would not be very well identified, due to their lack of sensitivity. In other words, we would get wider bounds for the acceptable parameters, if we applied the model only to the first 2 or 3 years. Many of these acceptable parameter sets

would then be considered unacceptable in the remaining years, since they would fail to predict correct responses in the very dry years that were not represented in the first period. A very small proportion of parameter sets would probably be acceptable for both periods. But to get a sufficient number of these acceptable parameterizations for both periods would require very many more simulations (which would just not be practicable). In summary, we can use the model to learn from the data in the most efficient way by applying the GLUE procedure for the entire 6-year period. In fact, split validation would almost certainly not be a good approach even if our time series had been much longer than 6 years. The inefficient and potentially highly misleading nature of the split-sample validation procedure has been demonstrated, for example, by Arsenault et al. (2018, Journal of Hydrology, 566, 346-362) who used a time series of 16 years and came to the conclusion that split validation is "… *futile at best, detrimental at worst, and deceiving in all cases*". In a similar study, Shen et al. (2022, Water Resources Research, doi:10.1029/2021WR031523) arrived at the same conclusion, stating that … "*calibrating to the full available data and skipping model validation entirely is the most robust split-sample decision*".

To answer the other question posed here: no, if the model we were using was inadequate we would not get good results from taking the best 30 simulations of 2000 parameterizations. We know this for sure because we have also run calibrations on these data sets for a model which is identical in all respects to the one described in our paper, except that it does not account for compensatory root water uptake (i.e. the Feddes root water uptake model was used instead). Even the best 30 simulations with this model at Selhausen are extremely poor.

Our main aim with the modelling was to investigate some likely plausible reasons for the measured differences in hydrological and plant response at two sites with the same soil type but contrasting climates. As the referee suggests, the (mostly) acceptable model efficiencies give us a lot of confidence that the model we used is good enough for this purpose.

However, we realize that our objectives with the modelling were not well explained at the end of the introduction at lines 118-126, as pointed out by the referee in a later comment. We have modified this part of the text to make the objective of the modelling exercise clearer. We now write:

*In this study, we make use of data from the TERENO-SoilCan network, in which large weighing lysimeters containing undisturbed soil monoliths have been transferred among several locations in Germany to emulate expected changes in climate (Zacharias et al., 2011; Pütz et al., 2016; Groh et al., 2020b). Here, we compare six years of measurements of the soil water balance and grassland production made in replicate lysimeters containing the same soil type, but located at two different sites with contrasting climates with simulations using a simple eco-hydrological model. Our main objective with this modelling exercise was to explore and identify some plausible mechanisms that would explain the observed responses of the grassland to a change in climate, in terms of biomass production and water use efficiency.*

The conclusion section contains some discussion but does not mention the interesting fact that plants were able to mitigate water stress in the drier climate by enhancing below ground (root) growth. If this is confirmed, soil organic matter contents could be expected to rise when climate change triggers more water stress. Also, in agricultural (grain) crops, above ground

yields might then be expected to diminish to favour belowground development. This is important in several ways, one of them being the expected yield of agricultural crops.

Yes, we certainly agree that this is an interesting and important result, but we did in fact already mention this in the conclusions (at lines 699-700). Related to the last part of this comment, it is worth mentioning that another study with the SoilCan lysimeters containing arable crops has shown the opposite effect i.e. that in the drier climate, water use efficiency increased, with larger grain yield per unit of water consumed. We suppose that putting more resources into seed production is probably a useful evolutionary adaptation for annual crops, whereas it seems that perennials try to survive droughts by enhancing root growth.

Scientific questions/issues:

l118-126 This final part of the introduction appears to refer to Materials and methods. I suggest finalizing the introduction with a concluding statement (possibly containing the objectives), and start the M&M section with the content of these lines.
Yes, we agree, and we have modified the text accordingly

Related to this, please state more clearly that the six lysimeters contain soil from Rollesbroich, but that three of them were transferred to Selhausen. l162 somehow contains this information, but it should be stated more clearly. The sentence in l179-181 is clear with this respect but should be located at an earlier position in the text.
This was actually stated already at lines 124-126 in the original manuscript. However, we have now re-written this text to make it even clearer.

l194-196 This net upward flow would imply a depletion of the groundwater body in the long term? I.e., if plants invest more in the belowground biomass to avoid drought stress, large-scale hydrological cycles would be affected? A comment in this respect seems appropriate here.

Possibly yes, but the Selhausen site is located on a low-lying relatively flat flood plain of the river Rhine (lines 130-131) and so it seems probable that this net upward flow is sustained by groundwater flow from the surrounding hills. In other words, the site lies in a discharge area in the landscape (rather than a recharge area). We have added some sentences to explain this (at lines 200-203 in the revised version):

*This is probably a result of the topographical position of the site on a low-lying flood plain, such that lateral groundwater flow from surrounding higher land is sufficient to maintain the supply of water to the drying plant root zone (i.e. the Selhausen site lies in a discharge area in the landscape).*

l210-213 Here you suggest that yield may have been affected by nutrient deficiencies. The performed simulations assume optimum plant nutrition (l356). Shortly discuss if this would affect the fairness of assessing model performance based on experimental data including nutrient issues.

*Yes, it would be more accurate to write that the model does not account for variations in plant nutritional status. We have modified the text at lines 371-372 accordingly:*

*… we assume that assimilation is limited by light, water and temperature, but not by variations in plant nutrition*

l284-294 No details are given about how you (numerically) solved the Richards equation (13). Was an implicit algorithm scheme used? How about the discretization? Time and space steps? Averaging K and h? Some more details would be important.

*Yes, we agree. We added these details at lines 318-323:*

*Equation 13 was solved by explicit finite differences and Runge-Kutta integration, with the soil profile divided into 25 numerical layers, with thicknesses varying from 1 cm (the uppermost layer) to 6 cm. A constant time step of 1 minute was employed to maintain numerical stability. The hydraulic conductivity regulating flow between two adjacent numerical layers in the soil profile was estimated by arithmetic averaging.*

l429 Regarding "the fraction of the total root length that is effective for water uptake" (epsilon), you use a fixed value of 0.05 (after Faria et al., 2010). A high model sensitivity to this parameter is to be expected, and it is also plausible to assume that epsilon will increase in periods of more intense root growth and decrease when the root system is shrinking due to stress. Please discuss shortly the effect of assuming a fixed static value for this parameter.

*Yes, our sensitivity analysis showed that the effective root fraction is indeed a moderately sensitive parameter (see table S2). However, for practical reasons, we could not include a large number of parameters in the uncertainty analysis. We selected four important parameters to calibrate and the reasons for this choice are already discussed at lines 518-533. One reason for not selecting the effective root fraction is that it would likely be correlated with other sensitive plant parameters, in particular with $f_{bg\ (opt)}$ and $\psi_{o(crit)}$.*

*This was described at line 533 in the original manuscript.*

l481-495 There are a lot of assumptions, guesses, and uncertainties in the soil hydraulic parameterization described here.
*Perhaps, but we believe that we have more data on which to base our parameterization than is usually the case in modelling studies of this kind.*

It is especially peculiar that the saturated water content of the surface layer is much higher in the lysimeters at Rollesbroich, as "estimated from the data by eye". Assuming all lysimeters contain a similar soil monolith (collected at Selhausen), can this be made plausible?
*Yes, we have added some text on this in the revised version at lines 517-523:*

*With only three replicates, it could be a result of chance spatial variation. However, at least two physical explanations appear plausible. It is possible that more optimal soil moisture conditions at Selhausen have led to faster mineralization rates of soil organic matter, leading*

*to a decline in the organic matter content and a concomitant increase in soil bulk density (i.e. a loss of porosity, Meurer et al., 2020). It may also be the case that the drier soil surface conditions at Selhausen have reduced soil wettability (Robinson et al., 2019).*

In l468-469, the reader is informed that a common parameterization for the soil hydraulic properties in all six lysimeters will be assumed. But apparently, this does not include ThetaS?
Yes, this is true. It is only common among the three replicates at each site. We have modified this statement accordingly (now at lines 490-491):

*…. assuming a common parameterization for the soil hydraulic properties in the replicate lysimeters at each site*

Please improve the description/justification of the soil hydraulic parameterization.
Yes, we have improved the text according to the above comments

l582 "No differences between the two sites were found for two of the parameters" – this is an interesting result and merits more attention. I think it is corroborated by common sense. Both parameters (the radiation extinction coefficient and the parameter controlling dry matter allocation and leaf loss as a function of water stress) are expected to be mostly genotypical and not affected by environmental conditions. On the other hand, unstressed stomatal conductance (discussed in l591-602) is expected to be a function of sink (atmosphere) conditions, especially VPD, temperature, radiation; rooting depth will probably be affected by soil moisture and temperature distribution in the soil profile, both a function of weather conditions.
Thank you for these helpful comments. We have modified the text (at lines 632-638 in the revised version of the paper) to reflect these ideas about the phenotypic plasticity of plant rooting and leaf conductance. In the latter case, we believe that it may be a response to heat stress, with increased transpiration rates cooling the plants. We now also cite a paper on this topic (Sadok et al., 2021) that was published after we submitted our paper.

*….. plants are known to acclimatize to environmental stresses by various physiological and morphological mechanisms (e.g. Nicotra et al., 2010; Tardieu et al., 2018; Vincent et al., 2020). For example, it is known that many plant species, including perennial ryegrass (Wedderburn et al., 2010), may respond to drought by developing deeper root systems. Although the mechanisms are still imperfectly understood, recent research suggests that various alterations in leaf physiology induced by heat stress may increase leaf hydraulic conductance, thereby enhancing transpiration rates and the degree of evaporative cooling (Sadok et al., 2021).*

l627 Here one of the reasons for the high E values at Rollesbroich is assumed to be "the capillary nature of the soil". This is a too vague description. Soils are very similar at both locations. I suggest you remove this reason.
Yes, we agree. We have deleted this.

l700-705 ("A major … growth") These statements are no conclusions from the presented research, they would fit better in the Discussion than in the Conclusion section.

We considered this option, but we prefer not to move this text, as we feel that it is a conclusion that we can draw from the results of our study.

Some other minor "technical corrections":

l62 SVAT stands for "Soil-Vegetation-Atmosphere Transfer". Please add "Transfer".'
Yes, OK, we have done so

l79 Johnson et al. (2008) would fit well in this list of citations (doi:10.1071/EA07133)
Yes, we have now cited this very relevant paper

l121 Delete "In this study".
Yes, O.K., we have done so

l235-240 For clarity, please add the units (dimensions) of the parameters of eqs. 1-7 in these lines.
Yes, we have done so. This was a useful exercise because we realized that a term was missing in equation 1, which ensures a correct conversion of units. We have fixed this.

l270 To be more precise, one would need pressures and conductances, both unavailable in bucket models.
Yes, we have added this

l288 (here and on other occasions) The unit "day" is officially abbreviated as d. I suggest using d instead of day, days throughout the text.
OK, we have done so

l306 I would prefer to say that tau (just like alpha and n) is a (fitting) shape parameter. The way you wrote it here, it seems tau could be independently measured.
We don't think that what we wrote implies anything about the way these parameters can be estimated. But we have now re-phrased the text at lines 317-318 to avoid misunderstanding:

$\alpha$ (m$^{-1}$) and n (-) are shape parameters and $\tau$ is a parameter that reflects the tortuosity and connectivity of the pore network.

l700 after "below-ground biomass" you might add: "thus mitigating drought stress".
Yes, we have done so

Figs. 1 and 2, X-axis label and Figs. 6, 7, 11, and 12, Y-axis: replace yr-1 by y-1
We prefer not to (but we will do so if the editor insists)

Fig. 4 Symbols in this figure can hardly be distinguished, I suggest using colours for the three lysimeters. It would also be good to clearly identify both columns of figures by Selhausen and Rollesbroich (they are now identified only by the codes of the respective lysimeters).
Yes, we have added the site names to the columns of figures. But adding colours to the symbols does not help. We tried this, but there is so much data (daily time resolution) and the replicates are so similar, that the symbols are indistinguishable regardless of how we plot this.

Figs. 4, 5, 8, 9, and 10: X-axis label is unclear. I would prefer "Time from onset of simulation (01-Jan-2013) [d]"

Yes, we agree that it was not ideal. But rather than change the x-axis label, we preferred to add some clarifying information in the caption i.e. Day 1 = 1$^{st}$ January 2013

Figure 7: Interpretation would be easier if Tp and Ta (and Ep and Ea) would appear in the same figure, side by side per location. In the current version of the figure, it is difficult to detect any difference between Tp and Ta.

Yes, we have done this (but the differences between Ta and Tp are indeed very small, as we discuss in the text in the following paragraph in relation to figure 8).

**Responses to referee 2**

**Major comments**

We would like to thank the referee for the many constructive comments on our manuscript, which have helped us to significantly improve the paper.

This is an interesting and very well written paper, that introduces an novel approach, that of transferring intact lysimeters from a wet cool climate to a much drier and somewhat warmer climate in Germany, to study the effects of climate (change) on the water balance and growth of grass vegetation. The paper has long introduction and materials and methods sections that already reveal a substantial amount of results and bring in a lot of literature references. At times it feels more like a review, and some of these references could perhaps have been saved for the discussion. The methods used are overall sound, although it was not fully clear to me why they had not selected a more recent and complete grass hydrology/growth model.

Some recent model validation exercises have suggested that existing grassland models have limitations with respect to descriptions of soil hydrological processes, especially with respect to water uptake during dry periods, something which was particularly relevant for our study. This was already noted in the paper at lines 83 to 96.

We had a rather specific purpose with the modelling. We built a relatively simple model to try to understand the reasons for differences in the water use efficiency and grassland growth caused by the move to a drier climate at the Selhausen site.

We made this objective clearer in the revised version. We now write:

*In this study, we make use of data from the TERENO-SoilCan network, in which large weighing lysimeters containing undisturbed soil monoliths have been transferred among several locations in Germany to emulate expected changes in climate (Zacharias et al., 2011; Pütz et al., 2016; Groh et al., 2020b). Here, we compare six years of measurements of the soil water balance and grassland production made in replicate lysimeters containing the same soil type, but located at two different sites with contrasting climates with simulations using a simple eco-hydrological model. Our main objective with this modelling exercise was to explore and identify some plausible mechanisms that would explain the observed responses of the grassland to a change in climate, in terms of biomass production and water use efficiency.*

Why was interception not included? This would not have caused much computational burden.

It didn't seem to be necessary to account for interception. As noted in the paper, the net evaporative loss caused by interception is usually quite small for short vegetation, since transpiration is reduced by a nearly equivalent amount.

What frustrated me a little was the fact the Results and Discussion section is rather short. After all this pre-amble on how the model works, the reader is still left with quite a few questions.

For example, what are the reasons for the upward flow in the drier lysimeters, even though LAI and Dry Matter are lower? Is this real? Is it the deeper root depth for these lysimeters that could have caused it. I would like to see the authors elaborate a bit more hear, for example by going back to the findings of their sensitivity analyses.

Yes, the net upward flow at Selhausen is definitely real (it has been measured). Yes, the deeper roots may contribute to this, together with the hydrogeological setting and the topographical position of the site on a low-lying flood plain. It seems likely that lateral groundwater flow from the surrounding higher land maintains the supply of water to the drying root zone.

We added some text to elaborate on this (at lines 200-203 in the revised version):

*This is probably a result of the topographical position of the site on a low-lying flood plain, such that lateral groundwater flow from surrounding higher land is sufficient to maintain the supply of water to the drying plant root zone (i.e. the Selhausen site lies in a discharge area in the landscape).*

**Minor comments**

Line 195-196: "It is also striking that the actual evapotranspiration slightly exceeds precipitation at Selhausen, so that the net percolation at the base of the lysimeters is negative (i.e. an upwards directed flow; Table 2)."…. Does this mean that the percolation was not measured separately? You say that the lysimeters enable the measurement of a complete (closed) water (line 113)? What is causing this negative percolation (capillary rise?).

Percolation was measured separately. We have clarified this in the revised version at lines 177-179:

*Water fluxes into and out of the lysimeters at the base are measured and are controlled by continuous measurements of pressure heads made in the surrounding soil at 1.4 m depth.*

The probable reasons for the negative net percolation are mentioned in our response above under "Major comments".

Chapter 2, Materials and Methods already contains a lot of results on the lysimeter water balance, AG biomass etc. Should that be moved to the results?

We preferred to keep this in the M&M section, as it sets the scene for the model development (why we model)

Line 214: water use efficiency (WUE) of the grassland in the drier climate was lower than that of the wet climate. Would we have expected the opposite (plants becoming more efficient under dry conditions)? So, is it really the leaf-level WUE hat has changed, or is this the result of other factors?

Yes, a lot depends on how WUE is defined. Note that we defined it (at line 184 in the original paper) from the data we have as the harvested yield divided by total evapotranspiration. The modelling that is described later explains the decrease in WUE as a consequence of an increased allocation of assimilates to the root system and a reduction in above-ground growth. So, this is not leaf-level WUE.

It also depends on the crop in question. We identified for arable lysimeters with a slightly different definition of WUE, that WUE increases after a move to a drier climate (see Groh et al. 2021: https://hess.copernicus.org/articles/24/1211/2020/

How exactly is infiltration calculated in the model?

This was explained at lines 292-297 in the original paper

And the flow at the bottom boundary of the soil profile?

This was explained at lines 290-292 in the original paper

What about Runoff?

There was no surface runoff, as the soil infiltration capacity was never exceeded. We have added a sentence in the revised version of the paper at lines 307-309 to explain this.

*It can be noted that it was not necessary to include surface runoff in the model because the soil infiltration capacity was never exceeded.*

In Line 424 and further you talk about feedbacks from the plant growth model to the hydrological model. You mention the effect of LAI and height on the aerodynamic resistances and hence on the ET fluxes. Surely LAI also affects radiation extinction, and therefore the energy available for ET. This could be mentioned too?

Yes, this is correct. We have clarified this in the revised version at lines 445-448:

*The root length density affects the soil resistance to water uptake by roots (equations 21 and 22), while the leaf area index affects both canopy and aerodynamic resistances (equations 8 and 10) as well as the interception of radiation by the canopy (equation 12).*

Also, the aerodynamic effects will have been relatively low. From that point of view would it not have been better to also consider interception (as it is also affected by LAI)? Although the values for grass are low, they are comparable to winter values of ET, and the equations required are straightforward?

It didn't seem to be necessary to account for interception. As the referee notes here, the net evaporative loss in interception is usually quite small for short vegetation.

Lines 489-492 You say: "The measurements from the matric potential sensors installed in the uppermost soil horizon (0-24 cm depth) appeared to be unreliable. We therefore also used the HYPRES pedotransfer functions to estimate the shape parameter n in the topsoil, while α

was set equal to the same value as the deeper horizons. First of all: why where these measurements unreliable? Was the soil too dry?

The pressure potential at 10 cm depth was measured by an MPS 1 sensor and not as in the other depths, by tensiometer. The MPS 1 sensor is relatively good at measuring pressure heads in dry soil, but in comparison with classical tensiometers, it is known to give unreliable values for the range between -200 cm until saturation. The problem here is that this range of pressure heads is important for the definition of the Mualem-van Genuchten parameters (especially theta s and alpha). Thus, in this study we didn´t used data from the MPS 1 sensor at 10 cm depth, and relied on the available tensiometer data.

How do you know that the deeper sensors could be deemed reliable?

This type of sensor was not installed in the deeper layers (only in the topsoil). We have made this clearer in the revised version at lines 179-182:

*Soil water contents and pressure heads are measured at a ten-minute time resolution at three depths (10, 30 and 50 cm depth) in the lysimeters using TDR probes and conventional tensiometers (30 and 50 cm depth) or MPS1 matric potential sensors (only at 10 cm depth).*

Also, can I ask why you did not use the VG parameters for the medium-layers where you did have measurements, instead of having to revert to generic HYPRES PTFs? Were these horizons too different?

Yes, that's right. There are large differences in texture between horizons (see lines 141-143 in the original version of the paper)

Your alpha parameter in Table 3 appears to be the same throughout the entire profile, yet n varies considerably. Is this realistic?

Yes, it does appear to be realistic (see figure S3 and table S1). The parameter *n* reflects the textural variations in the soil profile, whereas $\alpha$ is more closely related to soil structure.

How come that theta_s in the first soil layer is so much higher in Rollesbroich?

We have added text on this at lines 517-522 with two supporting references:

*With only three replicates, it could be a result of chance spatial variation. However, at least two physical explanations appear plausible. It is possible that more optimal soil moisture conditions at Selhausen have led to faster mineralization rates of soil organic matter, leading to a decline in the organic matter content and a concomitant increase in soil bulk density (i.e. a loss of porosity, Meurer et al., 2020). It may also be the case that the drier soil surface conditions at Selhausen have reduced soil wettability (Robinson et al., 2019).*

In Table 5 you talk about post-priori parameter ranges, whereas in the text (line 579-580) you mention that the posterior uncertainty ranges are much smaller than the prior uncertainty ranges. In figure 3 you talk about posterior distributions of the four parameters. Where exactly are you showing the prior uncertainty ranges? I find this all a little confusing.

We apologize for the confusion. We changed post-priori in table 5 to posterior. The prior uncertainty ranges are shown in the first column of table 5.

Lines 595-596: You say: "The simulations suggest that the maximum root depth at Selhausen has increased to ca. 80 cm, while the maximum stomatal conductance has roughly doubled". Were you not able to measure the root depth? I guess this would have caused destruction of the lysimeter core.

That's right. We would need to destroy the lysimeters to measure the root depth. This would not be in agreement with the goals of TERENO-SOILCan to provide long-term observations. However, we did observe the initial root depth at the time of sampling (see lines 147-149 in the original version of the paper)

Also, you say that stomatal conductance has doubled, but could it be that the parameter had assimilated aerodynamic effects to changes in vegetation structure? You hint at this perhaps in the following sentences, but it is not clear.

This seems unlikely. Aerodynamic resistance is calculated from plant height, which in turn is estimated from LAI. No differences in the relationships between plant height (and above-ground biomass) and LAI are apparent at the two sites (see figure S6), which suggests that the vegetation structure is similar.

In Figure 4 you need to make it clear which set of 3 graphs is representing which site (the same goes for Fig. 5). It is hidden in the legend but should be more explicit.

Yes, we have done this in the revised version

Also, based on these plots it is surprising that you opted for a theta_s of 0.55 for Selhausen (estimated "by eye"). I understand that these are daily averages (?), but during the winter months there much have been saturated conditions?

This is a misunderstanding. Theta_s was 0.45 at Selhausen not 0.55 (see table 3). Water contents do get close to saturation during winter, yes (see figure 4)

In Figure 5 it is quite hard to see what is going on. Would it be possible to separate the years somehow? Or perhaps make cumulative plots?

Yes, we have now included cumulative plots in the supplementary information.

Line 611-612: You say that the "model performs very well, matching the temporal dynamics in the high-time resolution data on state variables and fluxes as well as reproducing the differences in the overall water balances at the two sites". I am not sure that Table 6 reflects that statement? While model efficiencies are high the values for ET are much lower and values for LAI are negative (with LAI being a crucial variable in many equations), which makes me wonder whether Figure 5 hides a multitude of sins..?

Our statement in the paper only referred to the hydrological simulations (not the simulation of LAI) and Table 6 shows that the ME values are all positive for the hydrological variables.

Nevertheless, we have deleted this sentence as it is not really necessary. This statement was only intended as an introduction to the following text where we emphasize the importance of compensatory uptake mechanisms. We have re-written this section to make this clearer.

I find the discussion around Fig. 7 somewhat incomplete. While soil evaporation clearly depends on soil moisture content, windspeed etc. it also depends on incoming radiation (which would have been lower in Ro and higher in Se), and radiation reaching the soil through the canopy. Seeing the DM was lower in Se (and therefore LAI, see also figure 9) I would not necessarily have expected soil evaporation to have been lower for the Se site.

Yes, evaporation was apparently smaller at Selhausen despite higher radiation inputs. It must be due to dry surface soil. We have clarified this at lines 656-658:

*In contrast, soil evaporation is much smaller (ca. 50% of total evapotranspiration) in the drier climate at Selhausen despite greater incoming radiation, presumably because drying of the soil surface in summer frequently reduced evaporation below the potential rate (figure 7).*

Line 628: You say: At Rollesbroich" grassland is harvested 3-4 times during the growing season" Was this not the case at the Selhausen site? This makes comparison between the two sites difficult?

The grassland is also cut at Selhausen 3-4 times per season (see lines 150-153 in the original version of the paper)

The discussion around water stress, with a focus on Figure 8, seems to ignore the fact that one of your earlier figures indicated that the rooting depth at Selhausen was much deeper (80 cm) than at Rollesbroich. Is that mot the main reason for the relatively modest water stress experienced at this site?

Yes, this is good point. We have now included this in the discussion at lines 668-669:

*One reason for this is clearly the deeper root system.*

Line 692-693: Are some of these ME values less than excellent if the best value is 1? Can you provide a scale for what consititues poor, good, excellent etc. in the methods where you introduce ME?

Yes, we agree that this description is not really warranted. We changed "excellent" to "satisfactory". With ME, there is only one objective cut-off: an ME value of less than zero defines poor simulations (see line 559 in the original version of the paper).

**Responses to editor's comments**

Your manuscript was read by two reviewers, who were both positive about the manuscript, but also had some detailed and good points to still improve. Therefore, I suggest revisions with a further review by the reviewers.

Many thanks for your careful reading of the manuscript and your comments, which have helped us further improve the paper.

The response to the reviews seems well thought out and convincing. However, in several places the responses (often those to the second review) mainly answer the reviewers concern/ question, without clarifying whether / how the answer would be included in corresponding revision of the manuscript. In some of these cases I would think that further clarification in the manuscript would be good to avoid future readers from having the same doubts. For example the questions about the measurements of the matric potential, why were the measurements in the deeper layers reliable, but those of the topsoil not? The response is clear. But it is not clear whether you will clarify this in the manuscript.

We apologize for this oversight. It will now be clear where we have made changes

One of the reviewers comments which in my opinion needs some more consideration is the matter of the model performance. "The model performs very well, matching the temporal dynamics in the high-time resolution data on state variables and fluxes as well as reproducing the differences in the overall water balances at the two sites.". In the figures it is difficult to see more than that the general patterns of the moisture content and evapotranspiration are followed, but this high-time resolution is not visible at these scales. Also the model efficiencies for the deeper layer and ET are not very convincing. It is true that the model performs better than taking the average value for the whole time series as soon as the ME is above 0, but there is also a certain ME which one might expect any hydrological model to manage, based on the clear seasonality in hydrological data, isn't it? So the answer that the 4 to 6 and table 6 undeniably show that the statement is justified is not satisfactory.

Yes, we have deleted this statement in the paper. It was not really necessary and we do agree that the question of model performance and validity is really only worth discussing in a comparative sense … all models are (more or less) wrong, but some are less wrong than others!

I also think that the answer to the first reviewers point about the conclusions is rather short, you might want to clarify this a bit more: l700-705 ("A major… growth").

The referee wanted us to move this sentence from the conclusions to the discussion. We prefer to keep it where it is, as it is an important conclusion that we can draw from our study.

A final little point which caught my attention while looking through the article and figures was that the significance of the time/ site in the linear model analysis of Figure S2 seems strange. How can site not be significant in the abundance of herbs, but highly significant in the share of legumes? At a quick glance I would expect this differently..?

Thank you for pointing out this inconsistency. Previously, we did not scale the variables prior to ANOVA, which can lead to biased model fits if numerical values are strongly different among variables. We re-analyzed the ANOVA, centering the numerical variables, which resulted in the expected changes: a significant site effect for herbs and a slightly reduced effect for legumes. However, the overall main effects of a temporal change and different responses of grasses, herbs and legumes is not compromised by these adjustments.

Overall it is an interesting manuscript and I am looking forward to the new version, revised according to your replies to the reviewers with consideration of the comments above.

Kind regards,

Loes van Schaik